# Direct 1,3-butadiene biosynthesis in *Escherichia coli* via a tailored ferulic acid decarboxylase mutant

Yutaro Mori[1], Shuhei Noda[1], Tomokazu Shirai ⬤ [1✉] & Akihiko Kondo[1,2,3]

The C4 unsaturated compound 1,3-butadiene is an important monomer in synthetic rubber and engineering plastic production. However, microorganisms cannot directly produce 1,3-butadiene when glucose is used as a renewable carbon source via biological processes. In this study, we construct an artificial metabolic pathway for 1,3-butadiene production from glucose in *Escherichia coli* by combining the *cis,cis*-muconic acid (*cc*MA)-producing pathway together with tailored ferulic acid decarboxylase mutations. The rational design of the substrate-binding site of the enzyme by computational simulations improves *cc*MA decarboxylation and thus 1,3-butadiene production. We find that changing dissolved oxygen (DO) levels and controlling the pH are important factors for 1,3-butadiene production. Using DO–stat fed-batch fermentation, we produce $2.13 \pm 0.17\,g\,L^{-1}$ 1,3-butadiene. The results indicate that we can produce unnatural/nonbiological compounds from glucose as a renewable carbon source via a rational enzyme design strategy.

[1] Center for Sustainable Resource Science, RIKEN, Yokohama, Japan. [2] Department of Chemical Science and Engineering, Graduate School of Engineering, Kobe University, Kobe, Japan. [3] Graduate School of Science, Technology and Innovation, Kobe University, Kobe, Japan. ✉email: tomokazu.shirai@riken.jp

In efforts to meet the United Nations sustainable development goals to solve environmental problems, biorefinery processes in which renewable nonfood feedstocks are used as raw materials are being used instead of petrochemical refinery processes to prevent fossil fuel depletion. With recent advances in genome-editing technology, synthetic biology has achieved the ability to produce desired compounds via engineered microorganisms[1,2]. Many researchers have investigated the microbial production of valuable chemicals that are currently produced from petroleum, such as fuels, bulk chemicals, amino acids, and pharmaceuticals[3–8]. Many studies have aimed to improve target compound yields by enzyme screening and enzyme engineering[9]. Furthermore, in industrial biotechnology, fermentation and distillation/recovery processes are important factors, as is the development of microorganisms that produce target compounds. Although the development of synthetic biology has led to the biosynthesis of many compounds[10,11], biological production of novel unnatural/nonbiological compounds remains challenging.

1,3-Butadiene is a C4-conjugated diene with the simplest structure and is one of the most valuable industrial raw materials used for producing synthetic rubber (e.g., styrene-butadiene rubber and polybutadiene rubber) and engineering plastic (e.g., acrylonitrile-butadiene-styrene resin)[12]. The global market of 1,3-butadiene was $19 billion in 2019, and its global annual demand is >12 million metric tons. Nearly all 1,3-butadiene is typically synthesized from a C4 fraction that is a byproduct of naphtha cracking during ethylene production[13]. However, given that inexpensive ethylene derived from shale gas is currently produced, the supply of 1,3-butadiene has declined owing to the reduced production of ethylene derived from naphtha, and the gap between the supply and demand of 1,3-butadiene is predicted to increase[14]. To solve this problem and generate a sustainable society, there is a demand for switching from petroleum-based to biobased 1,3-butadiene.

In recent years, research on biobased 1,3-butadiene production has increased[15]. However, these are chemically synthesized products from biobased ethanol, 1- or 2-butanol, 1,3- or 2,3-butanediol, or tetrahydrofuran[16–18]. The existence of three different pathways have been suggested for 1,3-butadiene production from glucose via crotonyl-CoA, erythrose-4-phosphate, and malonyl-CoA[19], but the direct 1,3-butadiene biosynthesis from glucose as a renewable carbon source has not been achieved.

Muconic acid is an unsaturated six-carbon carboxylic acid with a conjugated diene and exists as three different isomers: *cis,cis*-muconic acid (*cc*MA), *cis,trans*-muconic acid (*ct*MA), and *trans, trans*-muconic acid (*tt*MA)[20]. *cc*MA is a natural metabolite involved in the decomposition pathway of aromatic compounds, such as benzoic acid derivatives, and can be produced via multiple pathways with different intermediate compounds[4,21–23]. Here, it was assumed that 1,3-butadiene can be produced by two decarboxylation reactions of *cc*MA (Fig. 1).

Ferulic acid decarboxylase (FDC), a member of the UbiD family enzymes, mediates the decarboxylation of phenylacrylic acid derivatives and converts them to terminal alkenes[24]. Cofactor prenylated flavin mononucleotide (prFMN) was recently discovered, and it was revealed that FDC-binding prFMN catalyzes the decarboxylation reaction[25]. prFMN is biosynthesized from FMN and dimethylallyl monophosphate by prenyltransferases encoded by the *ubiX* and *pad1* genes. It was revealed that UbiD family enzymes encoded by the *ubiD* and *fdc1* genes may be associated with prFMN as a cofactor[26]. prFMN-binding FDC can recognize not only aromatic compounds but also α,β-unsaturated carboxylic acids to produce terminal alkenes[27–36]. Therefore, this FDC was selected as a template enzyme for producing butadiene from the α,β-unsaturated dicarboxylic acid *cc*MA.

Directed enzyme evolution with random mutagenesis is a powerful method, but it can be time consuming[37–40]. Thus, to narrow the mutant library, rational enzyme design based on in silico calculations is attractive. Changing the substrate specificity and improving enzymatic activities against unnatural substrates are reported with this strategy[41–43].

In this study, we perform a rational enzyme design based on the affinity between FDC and *cc*MA and tailored FDC to synthesize 1,3-butadiene. We select FDCs derived from *Aspergillus niger* and *Saccharomyces cerevisiae* (*An*FDC and *Sc*FDC, respectively). The *cc*MA decarboxylation reaction is achieved via amino-acid substitution to capture *cc*MA at the FDC substrate-binding site while conserving the amino-acid residues involved in the decarboxylation reaction. Next, by combining the *cc*MA-producing pathway with tailored mutant FDCs in *Escherichia coli*, we construct an artificial metabolic pathway for 1,3-butadiene production from glucose. By balancing the aerobic phase for *cc*MA production and the microaerobic phase for 1,3-butadiene production, we succeed in the direct production of 1,3-butadiene from glucose.

## Results

**Designing *An*FDC mutants for 1,3-butadiene biosynthesis from *cc*MA.** *An*FDC was selected as a template decarboxylase for 1,3-butadiene production. Decarboxylation by *An*FDC requires prFMN, Q282, and R173, and the substrate specificity of *An*FDC is determined by the other surrounding residues in the substrate-binding site. The model of α-methyl cinnamic acid-bound *An*FDC showed that R173 interacts with the carboxylate$_{react.}$ group of α-methyl cinnamic acid (Fig. 2a). In the substrate-binding site of FDC, hydrophobic amino-acid residues (L185, I187, M283, I327, A331, Y394, F397, and L439) and the methyl group of the side chains of Thr323 and Thr395 clustered in the surroundings of the aromatic ring to stabilize and recognize an α-methyl cinnamic acid. A model of *An*FDC binding to *cc*MA was created using molecular operating environment (MOE) software. In the constructed FDC-*cc*MA-binding model, it was observed that Y394 formed a hydrogen bond with the carboxylate$_{opp.}$ group, which is opposite the carboxylate$_{react.}$ group of *cc*MA.

*cc*MA decarboxylation reactions were assumed to be facilitated by an amino-acid substitution at the FDC substrate-binding site. This substitution would result in the capture of the carboxylate$_{opp.}$ group while those amino-acid residues (R173 and E282) involved in the decarboxylation reaction would be conserved. We generated single-substitution mutants in silico and arranged them in order of affinity improvement factor against *cc*MA, after which we produced the top 25 designs of *An*FDC mutants and investigated the activity of those mutants (Fig. 2b). For measurement of enzymatic activity, we coexpressed UbiX derived from *E. coli* together with *An*FDC to synthesize prFMN. As a result of in vivo screening, wild-type (WT) the enzymatic activity of *An*FDCs was capable of producing 1,3-butadiene, indicating that WT *An*FDCs can recognize *cc*MA as a substrate and mediate the decarboxylation of both *cc*MA and pentadienoic acid (PA; a *cc*MA-decarboxylated compound). We observed marked improvement in the activity of the three types of T395-substituted *An*FDC mutants, and *An*FDC T395Q, in particular, yielded a 109 ± 5.26-fold increase in butadiene production compared to that of the WT. An interaction formed between the substituted amino-acid residues (H, Q, N) and the *cc*MA carboxylate$_{opp.}$ group in the FDC mutant-*cc*MA binding model demonstrated this improved activity. Furthermore, the carboxyl$_{opp.}$ group appeared to interact with Y394 and each of the substituted amino-acid residues (Supplementary Fig. 1). Taken together, these results showed that

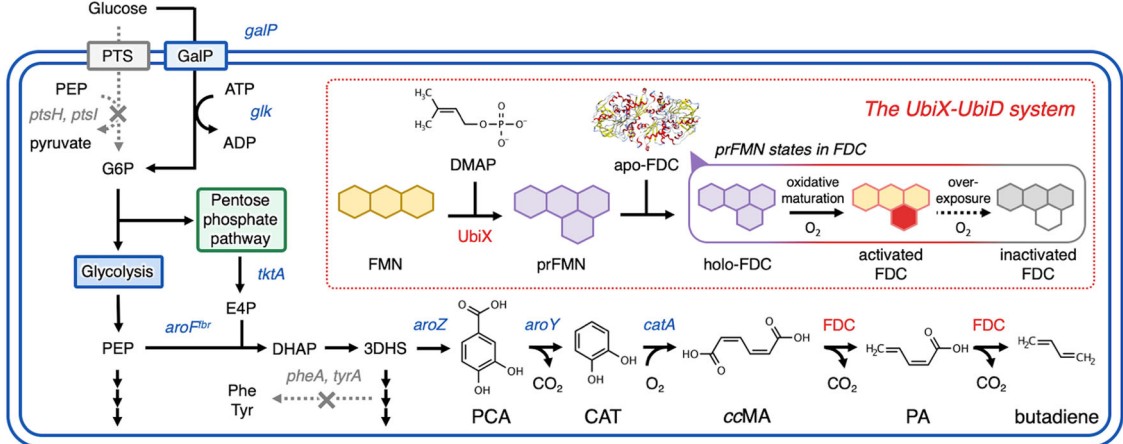

**Fig. 1 Artificial metabolic pathway for the direct production of 1,3-butadiene from glucose using the UbiX–UbiD system.** prFMN is biosynthesized via UbiX from both FMN and DMAP and then binds to apo-FDC. FDC requires exposure to oxygen for enzymatic activity, whereas overexposure to oxygen causes loss of activity. *PEP* phosphoenolpyruvate, *PTS* PEP-dependent phosphotransferase system, *ATP* adenosine triphosphate, *ADP* adenosine diphosphate, *G6P* glucose-6-phosphate, *E4P* erythrose-4-phosphate, *DAHP* 3-deoxy-D-heptulosonate-7-phosphate, *3DHS* 3-dehydroshikimate, *PCA* protocatechuic acid, *CAT* catechol, *ccMA* cis,cis-muconate, *PA* (Z)-pentadienoate, *Phe* L-phenylalanine, *Tyr* L-tyrosine, *FDC* ferulic acid decarboxylase, *FMN* flavin mononucleotide, *DMAP* dimethylallyl phosphate, *prFMN* prenylated FMN.

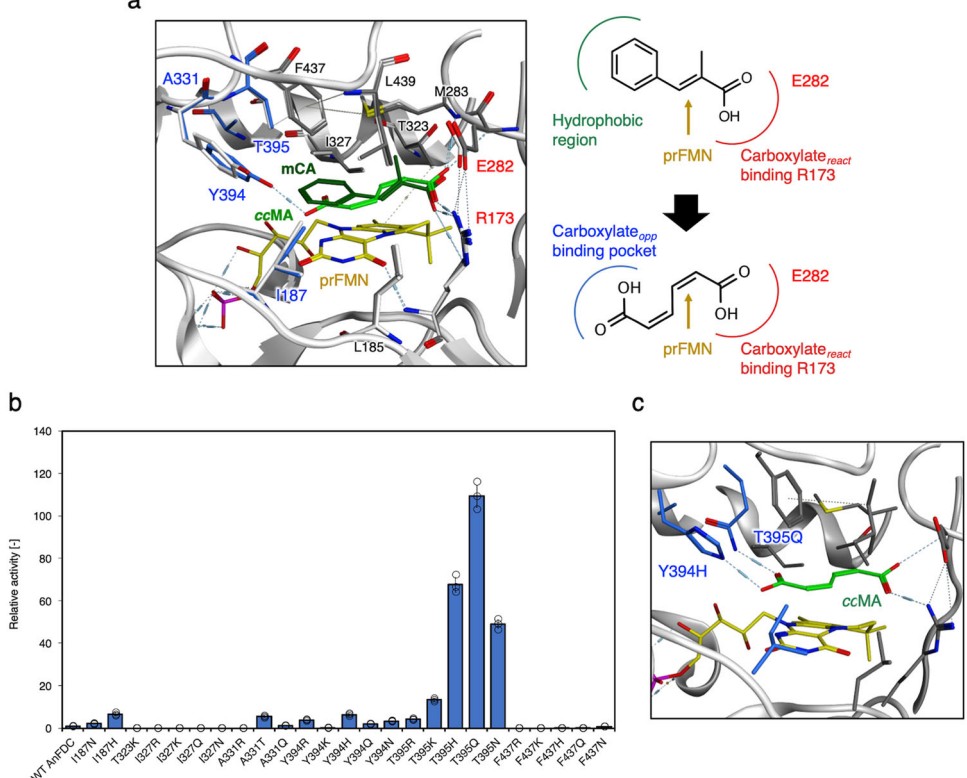

**Fig. 2 Design of *An*FDC for 1,3-butadiene production. a** Overlay of the active site of *An*FDC with bound α-methyl cinnamic acid (dark green) from PDB:4ZA7, a model of *cc*MA (light green)-bound *An*FDC with the lowest energy poses and a schematic design for *cc*MA. The negative hydrogen network and hydrophobic interactions are shown as dashed lines. **b** Relative decarboxylation activity of 25 *An*FDC mutants designed for *cc*MA. The activity of WT *An*FDC was defined as 1. The data are presented as the means ± SDs of three independent experiments (*n* = 3). **c** Simulated active site of the designed *An*FDC Y394H:T395Q with bound *cc*MA. *An*FDC ferulic acid decarboxylase derived from *Aspergillus niger*, *mCA* α-methyl cinnamic acid, *cc*MA cis,cis-muconic acid, *SD* standard deviation. Source data underlying Fig. 2b are provided as a Source Data file.

the T395 substitution site is a critical position for *cc*MA-*An*FDC binding.

To further improve substrate specificity for *cc*MA, we created a multiple-mutant library by combining these T395-substituted *An*FDC mutants and other single mutants that presented levels of activity higher than those of the WT (Supplementary Table 1). Compared with the WT *An*FDC, the best-designed *An*FDC Y394H:T395Q presented a 1002 ± 35.6-fold increase in 1,3-butadiene production from *cc*MA (Supplementary Table 2). The simulated model of *An*FDC Y394H:T395Q and *cc*MA showed

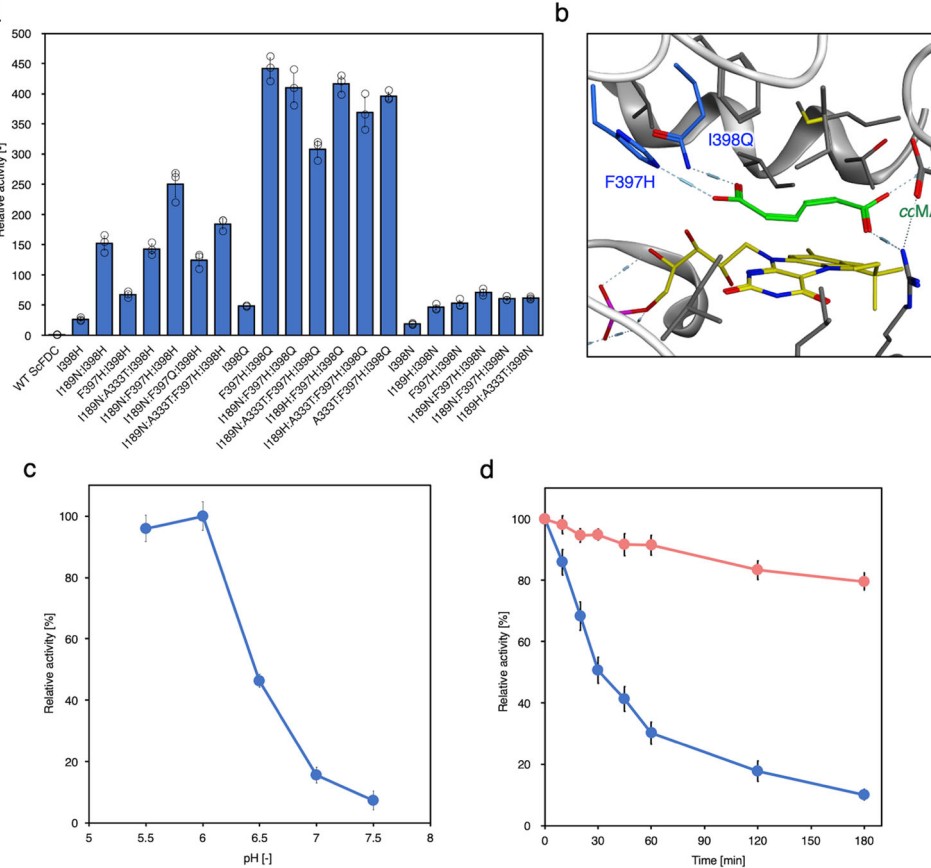

**Fig. 3 Design of *Sc*FDC for 1,3-butadiene production. a** Relative decarboxylation activity of the top 20 *Sc*FDCs designed for *cc*MA. The activity of WT *An*FDC was defined as 1. **b** Simulated active site of the designed *Sc*FDC F397H:I398Q with bound *cc*MA (based on PDB:4ZAC). The negative hydrogen network and hydrophobic interactions are shown as dashed lines. **c** pH dependence of the decarboxylation activity of the designed *Sc*FDC F397H:I398Q for 1,3-butadiene. **d** Time course of activity of *Sc*FDC F397H:I398Q under aerobic (blue) or oxygen-depleted (red) conditions. The activity was defined as 100% at 0 min after the incubation started. The data are presented as the means ± SDs of three independent experiments ($n = 3$). *Sc*FDC ferulic acid decarboxylase derived from *Saccharomyces cerevisiae*. *SD* standard deviation. Source data underlying Figs. 3a, 3c, and 3d are provided as a Source Data file.

that two substituted residues (H and Q) interact with the carboxylate$_{opp.}$ group of *cc*MA, maintaining the interaction between R173 and the carboxylate$_{react.}$ group of *cc*MA (Fig. 2c). Taken together, these results suggested that the interaction between the FDC mutant and *cc*MA enables the efficient capture of *cc*MA, improving the enzymatic ability to produce 1,3-butadiene.

**Developing and characterization of *Sc*FDC mutants by applying an *An*FDC mutant design.** We developed *Sc*FDC mutants by applying the good designs of the *An*FDC mutants. The sequence homology between the *An*FDC and *Sc*FDC proteins was 48.4%, and their substrate-binding sites have three differences; the amino-acid residues A331:Y394:T395 in *An*FDC correspond to V334:F397:I398 in *Sc*FDC (Supplementary Fig. 2). In addition to WT *An*FDC, WT *Sc*FDC also mediates *cc*MA decarboxylation (Fig. 3a). The best-designed *Sc*FDC F397H:I398Q, to which the best-designed *An*FDC Y394H:T395Q was applied, showed 441.9 ± 17.2-fold greater activity compared with that of WT *Sc*FDC. The simulated model of *Sc*FDC F397H:I398Q and *cc*MA showed that two substituted residues interact with the carboxylate$_{opp.}$ group of *cc*MA (Fig. 3b). Compared with *An*FDC Y394H:T395Q, *Sc*FDC F397H:I398Q-mediated *cc*MA decarboxylation more efficiently as the *cc*MA concentration increased and ultimately presented a 1.61 ± 0.35-fold increase in enzymatic activity (Supplementary Fig. 3).

The pH dependence of the enzymatic activity of *Sc*FDC F397H:I398 was analyzed. The optimal pH of *Sc*FDC F397H:I398Q for *cc*MA decarboxylation was 6.0, but the enzymatic activity significantly decreased at ~pH 7.0 (Fig. 3c).

Although AroY, which uses the same coenzyme that prFMN uses, requires exposure to oxygen to induce activity, it is known that overexposure to oxygen causes loss of enzyme activity[29]. Therefore, we also investigated the effects of oxygen on *Sc*FDC F397H:I398Q. After incubation of *Sc*FDC F397H:I398Q under aerobic or oxygen-depleted conditions, the *cc*MA was added and its enzyme activity was measured. The relationship between incubation time and the enzymatic activity of FDC is shown in Fig. 3d. The enzymatic activity of *Sc*FDC F397H:I398Q at 0 min after incubation started was defined as 100%. The *Sc*FDC F397H:I398Q activity has a half-life of 30 min under aerobic conditions but decreased to 10.1 ± 1.7% after 180 min. On the other hand, 79.6 ± 2.9% of the activity remained after 180 min under oxygen-depleted conditions. These results indicated that the activity of FDC decreased due to continued oxygen exposure.

We investigated the substrate specificity of *Sc*FDC F397H:I398Q using the *cc*MA isomers *ct*MA and *tt*MA. We observed that WT *Sc*FDC recognized both *cc*MA isomers as substrates and converted them to butadiene via two decarboxylation reactions. Although *Sc*FDC F397H:I398Q was designed for *cc*MA, 45.8 ± 4.2-fold (*ct*MA) and 12.0 ± 1.4-fold (*tt*MA) enhancement of butadiene production by *Sc*FDC F397H:I398Q was observed, compared with that of WT *Sc*FDC. These findings demonstrated

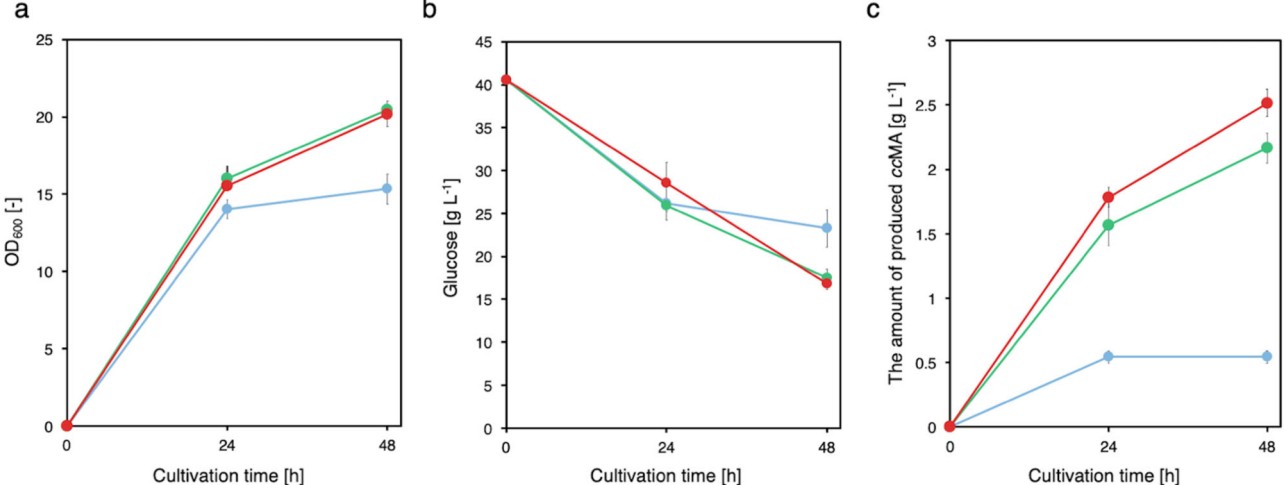

**Fig. 4 Culture profiles of *cc*MA-producing CFB01, CFB11, and CFB21 strains.** Time courses of **a** bacterial cell growth and **b** glucose consumption. **c** *cc*MA yield with cultivation time. Solid blue circles, CFB01; solid green triangles, CFB11; solid red squares, CFB21. The data are presented as the means ± SDs of three independent experiments (*n* = 3). *cc*MA *cis,cis*-muconic acid, *SD* standard deviation. Source data are provided as a Source Data file.

that the mutated residues interact with the carboxyl group$_{opp}$ of *cc*MA, promoting the capture of the carboxyl group$_{opp}$ of *cc*MA isomers, and that *Sc*FDC F397H:I398Q efficiently converted them to butadiene.

We investigated the substrate specificity of *Sc*FDC F397H:I398Q for PA, an intermediate reaction that occurs when muconic acid undergoes a single decarboxylation. The results showed that the relative activity of *Sc*FDC F397H:I398Q for WT *Sc*FDC was 1.12 ± 0.11. A *Sc*FDC F397H:I398Q and PA docking model is shown in Supplementary Fig. 4a. We confirmed that substituted F397H, I398Q, and a terminal alkene group of PA were separated. Furthermore, we investigated butadiene production from *cc*MA by combining WT *Sc*FDC and *Sc*FDC F397H:I398Q. The activity decreased to 56.3 ± 7.6% when WT *Sc*FDC and *Sc*FDC F397H:I398Q were coexpressed compared to that when only *Sc*FDC F397H:I398Q was expressed (Supplementary Fig. 4b). Based on these results, it was shown that *Sc*FDC F397H:I398Q can recognize both *cc*MA and PA as substrates and can efficiently produce 1,3-butadiene from *cc*MA by a double decarboxylation reaction; thus, only *Sc*FDC F397H:I398Q was used for the butadiene production pathway to be introduced into *E. coli*.

**Developing a *cc*MA-producing *E. coli* strain from glucose.** To construct an artificial pathway through which 1,3-butadiene is produced from glucose, we developed an *E. coli* strain for *cc*MA production. In this study, we selected the pathway involving protocatechuic acid (PCA), the shortest pathway reported to be a *cc*MA production pathway, and conducted genome engineering for efficient *cc*MA production (Fig. 1)[4]. In this *cc*MA pathway via PCA, *cc*MA is produced from 3-dehydroshikimic acid (3DHS) by 3DHS dehydratase (*aroZ*), protocatechuic acid decarboxylase (*aroY*), and catechol dioxygenase (*catA*) in a three-step reaction. We selected *aroZ* derived from *Bacillus thuringiensis*, *aroY* from *Klebsiella pneumoniae*, and *catA* from *Pseudomonas putida* DOT-T1E. For *cc*MA production, these genes were incorporated into CFB01, CFB11, and CFB21; these genes constitute the basis of aromatic derivative-producing *E. coli* strains (Fig. 4)[21,23]. Cell growth and *cc*MA production of CFB01 nearly stopped after 24 h of culture, and 0.54 ± 0.04 g L$^{-1}$ *cc*MA was produced. CFB11 and CFB21, the engineered strains designed for *cc*MA production, continued to grow and produce *cc*MA after 24 h of culture. After 48 h of culture, 2.17 ± 0.09 and 2.52 ± 0.08 g L$^{-1}$ *cc*MA were

produced by CFB01 and CFB02, respectively. Accumulations of intermediate metabolites, PCA, and catechol (CAT), were not observed. The highest yield of *cc*MA was 0.134 ± 0.054 mol (mol glucose)$^{-1}$, which occurred from the CFB21 culture (Supplementary Table 3). Taken together, these results showed that we successfully generated a *cc*MA-producing *E. coli* strain (CFB21).

**Constructing an artificial metabolic pathway for 1,3-butadiene production.** By combining the *cc*MA-producing *E. coli* strain from glucose with the *Sc*FDC mutant, we constructed an artificial pathway through which 1,3-butadiene is produced from glucose. CFB22, which is CFB21 harboring *UbiX*, was cultured, and its *cc*MA production slightly decreased (~10%) (Supplementary Fig. 5).

Next, we developed a 1,3-butadiene-producing strain by introducing *Sc*FDC into CFB21 and CFB22. These strains were cultured under aerobic conditions (24 h) and then packed culture (next 24 h) conditions. CFB222, which is the CFB22 strain with *Sc*FDC F397H:I398Q, grew and consumed glucose after packing (Supplementary Fig. 6). With this strain, 1.27 ± 0.03 g L$^{-1}$ *cc*MA and 1.07 ± 0.05 g L$^{-1}$ PA were produced (Fig. 5). A total of 41.4 ± 2.82 mg L$^{-1}$ of 1,3-butadiene was produced by CFB222. Moreover, CFB212, a strain in which *UbiX* was not overexpressed, produced no 1,3-butadiene but produced a small amount of PA. These results showed that *UbiX* overexpression is required for 1,3-butadiene production, although *UbiX* is inherently present in *E. coli*.

CFB222 was cultured in a closed 2-L medium bottle to examine the effects of oxygen conditions on butadiene production. FDC enzymatic activity requires oxygen for activation;[34] however, increased oxygen exposure leads to inactivation (Fig. 3d). In addition, *catA* requires oxygen to produce *cc*MA from CAT. Therefore, in this pathway for 1,3-butadiene production, aerobic conditions are suitable for the activation of FDC and for *cc*MA production in the early stage of culture, whereas oxygen-depleted conditions are needed to maintain FDC activity. In a closed bottle, the gas phase was first applied under aerobic conditions and then gradually under oxygen-depleted conditions as the cells grew and consumed oxygen. In this packed culture, CFB222 was cultured at medium:air phase ratios of 1:5, 1:10, and 1:20. Glucose consumption and cell growth increased with increasing medium: air phase ratio, although they stopped after 24 h of culture (Fig. 6a). Similarly, *cc*MA, PA, and butadiene yield also increased

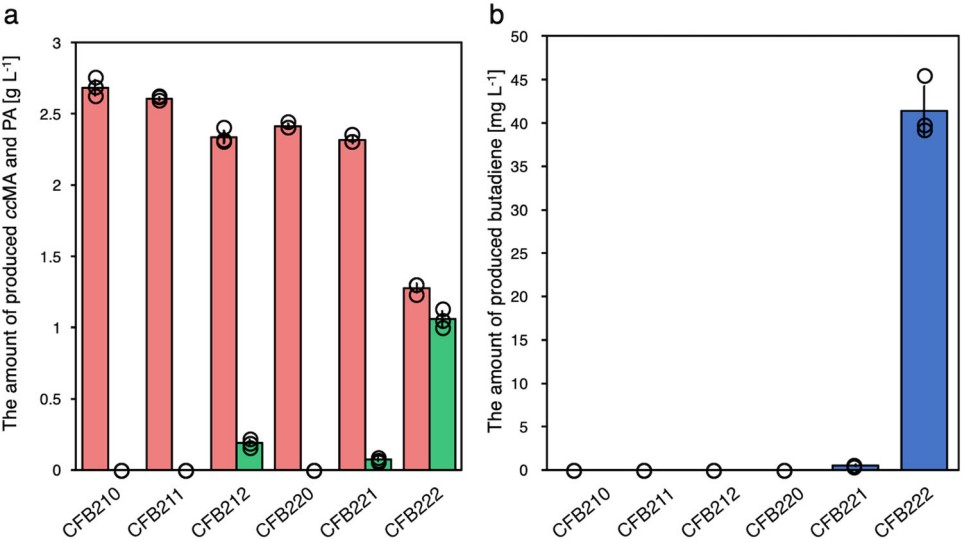

**Fig. 5 Culture profiles of 1,3-butadiene-producing CFB210–CFB222 strains under aerobic conditions (24 h) and packed cultivation (another 24 h) conditions. a** ccMA (red) and PA (green) and **b** butadiene (blue) yields after 48 h of cultivation. The data are presented as the means ± SDs of three independent experiments (n = 3). ccMA cis,cis-muconic acid, PA pentadienoic acid, SD standard deviation. Source data are provided as a Source Data file.

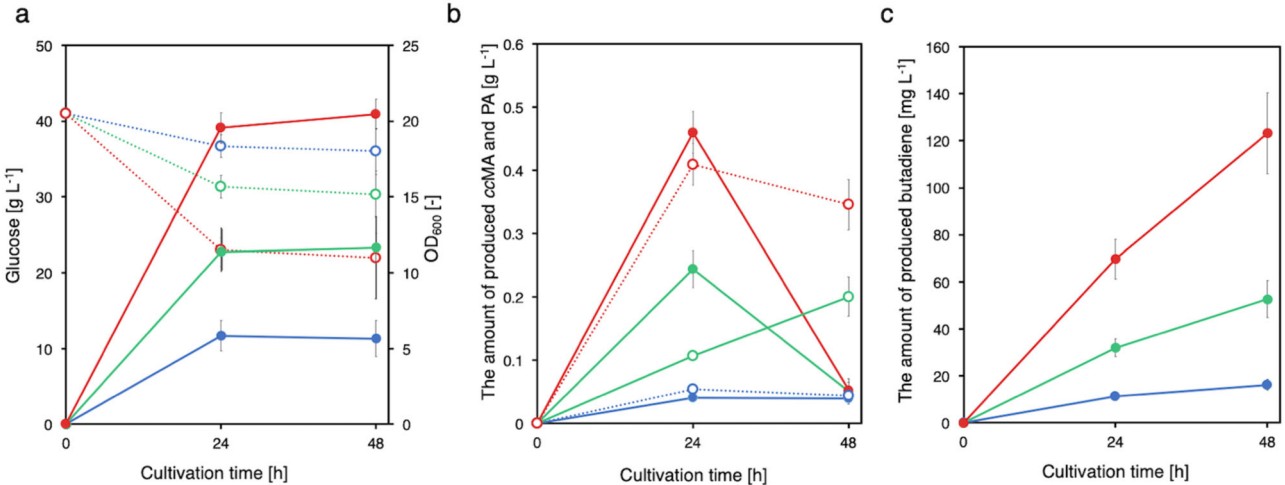

**Fig. 6 Culture profiles of 1,3-butadiene-producing CFB222 strains under microaerobic conditions. a** Time course of cell growth (solid circles, solid lines) and glucose consumption (open circles, dashed lines). The medium:air phase ratios were 1:5 (blue), 1:10 (green), and 1:20 (red). **b** Time courses of ccMA (solid circles, solid lines) and PA (open circles, dashed lines) produced. The medium:air phase ratios were 1:5 (blue), 1:10 (green), and 1:20 (red). **c** 1,3-Butadiene yield with cultivation time. The medium:air phase ratios were 1:5 (solid blue), 1:10 (solid green), and 1:20 (solid red). The data are presented as the means ± SDs of three independent experiments (n = 3). ccMA cis,cis-muconic acid, PA pentadienoic acid, SD standard deviation. Source data are provided as a Source Data file.

with an increase in the medium:air phase ratio (Figs. 6b and 6c). These results suggested that balancing aerobic and oxygen-depleted conditions in culture is an important factor for 1,3-butadiene production. Culturing (during which the pH was adjusted) was subsequently performed to examine the effects of pH on butadiene production (Supplementary Fig. 7). Butadiene production was higher at a low pH than at a pH of approximately 7.0, which is optimal for *E. coli* growth.

Based on the mass spectrum of standard 1,3-butadiene (m/z = 50–54), the percentage of 1,3-butadiene isotopes produced was calculated (Supplementary Fig. 8). The main peak of 1,3-butadiene produced from natural glucose occurred at m/z = 54 (Supplementary Fig. 9). With [U-$^{13}$C] glucose, a specific peak of [$^{13}$C$_4$] 1,3-butadiene at m/z = 58 (>70%) was observed, and peaks of [$^{13}$C$_{0-3}$] 1,3-butadiene were also detected (Supplementary Fig. 10). It was suggested that [$^{13}$C$_{0-3}$] 1,3-butadiene was produced

from [U-$^{13}$C] glucose and carbohydrates in a yeast extract of the medium. When the yeast extract in the medium was reduced, the specific peak derived from [$^{13}$C$_4$] 1,3-butadiene increased. Finally, in the absence of a yeast extract, [$^{13}$C$_4$] 1,3-butadiene was predominant (>95%). Taken together, these results indicated that 1,3-butadiene was produced mainly from glucose.

**Production of 1,3-butadiene from glucose in a jar fermenter.** Jar fermentation was performed under three different dissolved oxygen (DO) conditions at a pH of 7.0 (Supplementary Fig. 11). Under aerobic conditions (Supplementary Fig. 11a–c), the concentration of ccMA reached 3.28 ± 0.29 g L$^{-1}$ at 48 h. Although the production of 1,3-butadiene stopped at 60 h, 123 ± 8 mg L$^{-1}$ 1,3-butadiene was produced at 96 h. Under microaerobic conditions (Supplementary Fig. 11d–f), the culture profile was almost the

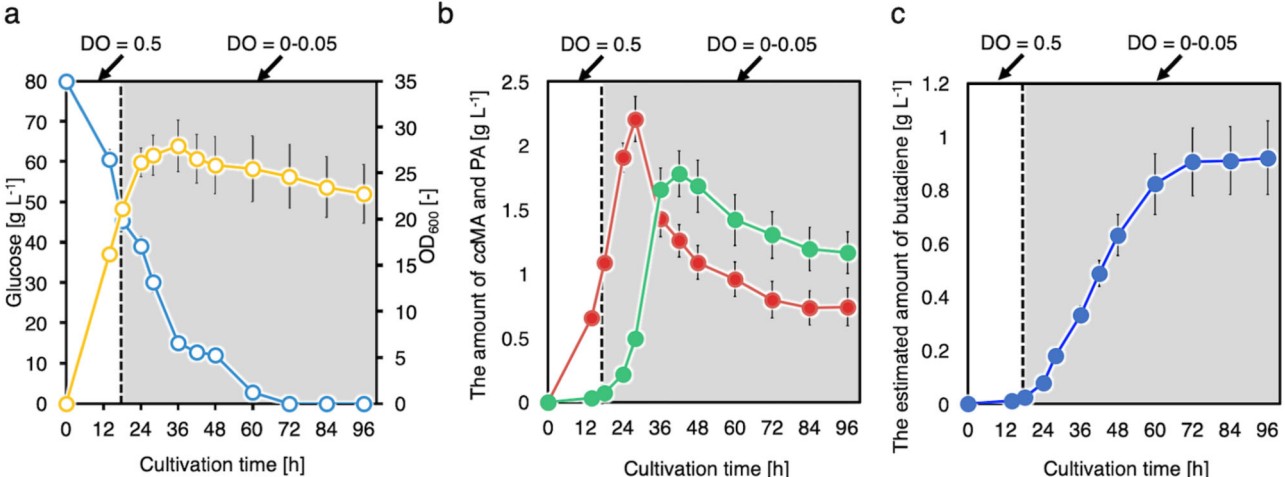

**Fig. 7 Batch culture of CFB222 in a 1-L jar fermenter under DO-switching conditions at a pH of 6.0.** Time courses of **a** bacterial cell growth (open yellow circles) and glucose concentration (open blue circles), **b** $ccMA$ (solid red squares) and PA (solid green triangles) yields, and **c** estimated butadiene yield (solid blue circles). The data are presented as the means ± SDs of three independent experiments ($n = 3$). DO dissolved oxygen, $ccMA$ $cis,cis$-muconic acid, PA pentadienoic acid, SD standard deviation. Source data are provided as a Source Data file.

same as that under aerobic conditions. These results suggested that a sufficient amount of $ccMA$ was supplied for butadiene production even under microaerobic conditions. Under DO-shifted conditions (from aerobic to microaerobic conditions at 18 h) (Supplementary Fig. 11g–i), 1,3-butadiene production increased to $260 ± 20 \, mg \, L^{-1}$ compared with that of the two above-mentioned conditions, although intermediate compounds ($3.0 ± 0.27 \, g \, L^{-1}$ $ccMA$ and $1.5 ± 0.16 \, g \, L^{-1}$ PA) accumulated. Therefore, we noted that switching from aerobic to microaerobic conditions is required when aiming to increase butadiene production.

Next, CFB222 fermentation under DO-shifted conditions (from aerobic to microaerobic conditions at 18 h) at a pH of 6.0 was performed (Fig. 7). Glucose consumption and cell growth were slower than they were at a pH of 7.0 (Supplementary Fig. 11g). Although the $OD_{600}$ value at 18 h was $27.3 ± 1.3$ and reached $28.9 ± 1.9$ at a pH of 7.0, the $OD_{600}$ value at 18 h was $21.2 ± 0.8$ and reached $28.0 ± 0.7$ at a pH of 6.0. In addition, 1,3-butadiene was continually produced until 96 h. Although $0.74 ± 0.13 \, g \, L^{-1}$ $ccMA$ and $1.17 ± 0.18 \, g \, L^{-1}$ PA accumulated at the end of the culture, $0.91 ± 0.09 \, g \, L^{-1}$ 1,3-butadiene was produced. Therefore, culturing at a pH of 6.0 resulted in higher butadiene production than did culturing at a pH of 7.0.

To further increase butadiene production, DO-stat fed-batch fermentation was performed (Fig. 8). Cell growth increased, and the maximum $OD_{600}$ reached $35.0 ± 3.2$. 1,3-Butadiene was produced until the end of the culture, although $1.37 ± 0.06 \, g \, L^{-1}$ $ccMA$ and $2.33 ± 0.41 \, g \, L^{-1}$ PA accumulated. In total, $2.13 ± 0.17 \, g \, L^{-1}$ 1,3-butadiene was ultimately produced from glucose at 96 h. We succeeded in increasing 1,3-butadiene production significantly via DO-stat fed-batch fermentation by changing the level of DO at a low pH.

## Discussion

1,3-Butadiene is a C4 diene and one of the most valuable industrial compounds. In this study, based on a rational design strategy of protein mutagenesis for developing target enzymatic functions, we constructed the artificial metabolic pathway for direct 1,3-butadiene production from glucose, which involves an entirely biological process.

Multiple classes of enzymes can produce biochemical compounds that have a terminal double bond. Fatty acid and fatty aldehyde decarboxylases have widespread substrate specificities and produce

C4–C20 terminal alkenes, which are important for sustainable fuel compounds[44]. Some dehydratases can also produce alkenes and acrylic acids by dehydration of the near-terminal hydroxyl group of substrates[45,46]. In a recent study, a class of enzymes was discovered: the biochemical reactions of these enzymes result in the formation of a terminal double bond, releasing ammonia, and formaldehyde[47,48]. The FDC used in this study, which contains prFMN biosynthesized by prenyltransferases, can catalyze the decarboxylation of $\alpha,\beta$-unsaturated carboxylic acids.

Computer-aided rational enzyme design can help improve the desired functions of target proteins, such as thermostability, enzymatic activity, and substrate specificity. With the evolution of computer simulation technology, many studies of enzyme development based on calculated parameter changes of substrate-docked enzyme modeling before and after in silico mutation have been reported[49–54]. In this study, the best design, $An$FDC Y394H:T395Q, achieved a $1002 ± 35.6$-fold improvement in butadiene-producing activity compared with that of WT $An$FDC. The computer-simulated model of $An$FDC Y394H:T395Q shows that Y394H and T395Q interact with the carboxyl$_{opp}$ group of $ccMA$ while maintaining the interaction between R173 and the carboxyl$_{react}$ group. In addition, $Sc$FDC F397H:I398Q, to which $An$FDC Y394H:T395Q was applied, presented a $441.9 ± 17.2$-fold increase in activity compared with that of WT $Sc$FDC. Similarly, $Sc$FDC F397H:I398Q also interacted with $ccMA$, suggesting that this interaction altered substrate specificity and thereby improved the affinity of the FDC mutant for $ccMA$ and increased enzymatic activity.

The optimal pH for the $ccMA$ decarboxylation reaction involving $Sc$FDC F397H:I398Q is 6.0, with a substantial decrease in activity at ~pH 7.0. In contrast, the optimal pH for the decarboxylation activity of WT $Sc$FDC against the natural substrate cinnamic acid was 7.0, and the enzymes could maintain >90% of its activity at the pH range of 6.0–7.0[55]. Considering a pKa of the substituted His residue of ~6.0, we assumed that the protonated form of the imidazole side chain of His is required for efficient binding of $ccMA$ to $Sc$FDC F397H:I398Q. AroY, an enzyme of the UbiD family, contains prFMN and is known to catalyze the decarboxylation of protocatecuate. The enzymatic activity of UbiD species requires oxygen to form the active prFMN$^{iminium}$ species[30,35], however, increased oxygen exposure leads to decrease the activity of some Ibid-Family members such as AroY[29]. It is reported that oxidative maturation of prFMN is required for FDC

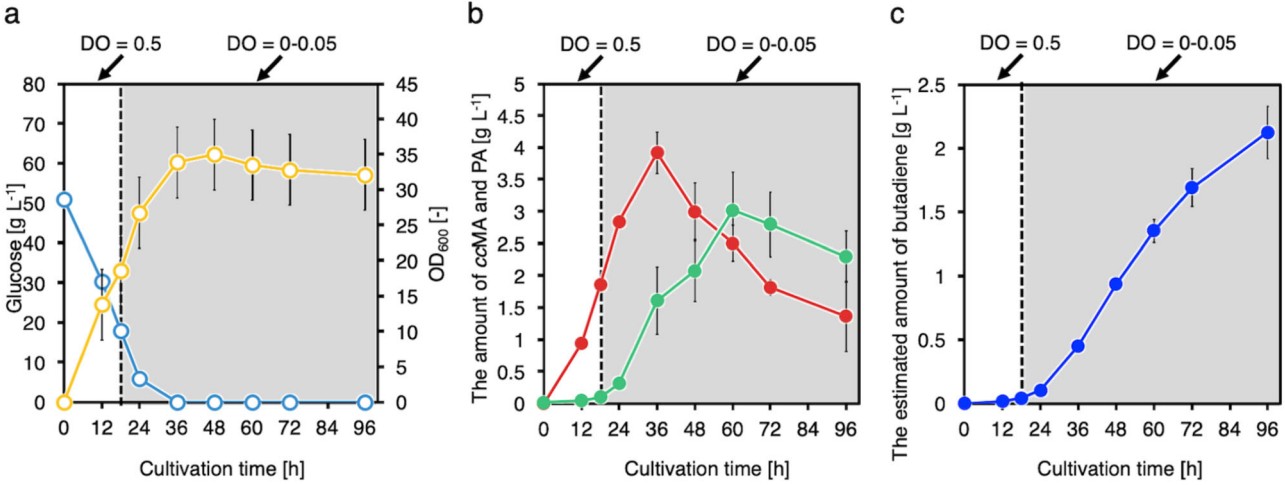

**Fig. 8 DO-stat fed-batch cultivation of CFB222 in a 1-L jar fermenter under DO-switching conditions at pH 6.0.** Time courses of **a** bacterial cell growth (open yellow circles) and glucose concentration (open blue circles), **b** ccMA (solid red squares) and PA (solid green triangles) yields, and **c** estimated butadiene yield (solid blue circles). The data are presented as the means ± SDs of three independent experiments ($n = 3$). DO dissolved oxygen, ccMA cis, cis-muconic acid, PA pentadienoic acid, SD standard deviation. Source data are provided as a Source Data file.

activity[34], although no oxygen sensitivity of the active holo-enzyme has been reported. The oxygen sensitivity of purified ScFDC F397H:I398Q was measured in vitro, and a decrease of enzymatic activity of FDC under aerobic conditions was observed. To produce butadiene from ccMA, the two decarbox-ylation reactions by ScFDC F397H:I398Q are required, thus butadiene production may decrease drastically under aerobic conditions. Perhaps, the conformational change induced by ScFDC F397H:I398Q mutations in this study may lead to $O_2$ sensitivity not observed for the WT ScFDC. The difference in decarboxylase activity between AnFDC Y394H:T395Q and ScFDC F397H:I398Q may depend on the differences in the ratio of the active/inactive prFMN contained[25,27,34]. ScFDC F397: I398Q activity for PA was largely unchanged compared with that of the WT ($1.12 ± 0.11$-fold increase). The PA molecule is smaller than muconic acid is. Therefore, we assumed that when PA forms a reactive binding state, it is less likely to interact with the sub-stituted amino-acid residue F397H:I398Q and would not affect the affinity for PA. When WT ScFDC and ScFDC F397H:I398Q coexisted, the activity on ccMA decreased ($56.3 ± 7.6\%$) compared with that when only ScFDC F397H:I398Q was present. Although the activity for PA was virtually the same for WT ScFDC and ScFDC F397H:I398Q, ScFDC F397H:I398Q presented extremely high activity for ccMA. Therefore, we assume that the first stage of the reaction from ccMA to PA decreased when WT ScFDC and ScFDC F397H:I398Q coexisted compared with that when only ScFDC F397H:I398Q was expressed.

We constructed the artificial metabolic pathway for 1,3-buta-diene from glucose by combining a ccMA-producing pathway with the decarboxylation of ccMA by FDC. The butadiene-producing reaction in this study involves a two-step reaction that uses ccMA as an initial substrate, with PA generated from ccMA (first step) and butadiene generated from PA (second step) by decarboxylation reactions. The standard Gibbs free energy of the reaction is $\Delta G^{0'} = -14.2 \text{ kJ mol}^{-1}$ for both reactions, making it thermodynamically[56–58]. At room temperature, butadiene is gaseous and insoluble in water; thus, the second step of the reaction is associated with a strong positive value. This reaction is therefore considered extremely useful as an artificial metabolic pathway for butadiene production.

Oxygen is required for prFMN activation and ccMA synthesis by catA, whereas oxygen-depleted conditions are needed to

maintain FDC activity. This was verified by the change in buta-diene yield in an experiment in which packed bottles with dif-ferent medium:air phase ratios were used. These results prompted us to change the DO conditions from being aerobic to micro-aerobic during culturing, and as a result, we successfully increased the butadiene yield. AroY, which generates CAT from PCA in the ccMA production pathway, is an enzyme that associates with prFMN as a coenzyme; therefore, we assumed that AroY activity would be affected by UbiX expression in the same way as FDC was. However, in vitro culturing of ccMA-producing bacteria did not result in an increase in ccMA production regardless of UbiX expression. This may have occurred because AroY was suffi-ciently active to produce ccMA owing to prFMN produced by endogenous UbiX. Although it is possible that AroY activity is reduced by oxygen, ccMA was produced by AroY and also by CatA in an experiment in which test tubes and jars were used under aerobic conditions, demonstrating that AroY has sufficient activity under these experimental conditions.

Jar fermentation at a relatively low pH increased 1,3-butadiene production. We propose two possible explanations for this. (1) The cell growth rate was lower at a pH of 6.0 than at a pH of 7.0, and proliferation of its growth continued after the conditions were shifted to microaerobic conditions. Therefore, we assume that an active form of the ScFDC mutant was produced that increased butadiene production. (2) The calculated $pKa$ values are 4.89 for PA and 3.87 and 4.65 for ccMA, indicating that the proportion of PA and ccMA protonated forms increases under low pH conditions. The protonated forms of ccMA and PA can reenter cells in a medium, which may promote the conversion to butadiene. The necessity of switching culture conditions from aerobic to microaerobic conditions and the requirement of a low pH have been shown previously. We therefore applied this knowledge to DO-stat-fed-batch cultures and thereby successfully increased butadiene production.

Two decarboxylation reactions are required for 1,3-butadiene production; that is, two moles of $CO_2$ are lost, which results in a low carbon yield. Several recent studies on direct $CO_2$ fixation involving the use of E. coli have been reported[59,60]. When this technique is applied to 1,3-butadiene-producing strains to fix $CO_2$, increased carbon yield from 1,3-butadiene production is expected.

The intermediate substrates ccMA and PA remained in the medium at the end of fermentation. This means that the activity

of the developed *Sc*FDC mutant is still insufficient. Therefore, we hypothesize that butadiene production may be substantially increased when a mutant with increased affinity for PA is used and when that mutant is combined with *Sc*FDC F397H:I398Q, as developed in this study. FDC mutant enzyme activity depends on the pH, of which the optimum is 6.0, and enzymatic activity may be considerably reduced when the *E. coli* intracellular pH is 6.8–7.5[61]. Therefore, we assumed that changing the optimal pH of FDC mutants to 6.8–7.5 would promote FDC activity in *E. coli* and butadiene production.

Based on the rational enzyme design of the substrate-binding site, FDC was tailored to recognize the unnatural substrate *cc*MA and efficiently produce 1,3-butadiene via two decarboxylation reactions. By combining the *cc*MA production pathway and the decarboxylation of FDC, we generated an artificial metabolic pathway for direct 1,3-butadiene production. For 1,3-butadiene production, controlling the DO and pH is critical, and by optimizing these parameters, we achieved a dramatic increase in 1,3-butadiene yield. Because of the cellular toxicity of the intermediate compound PA, a tailored FDC mutant for PA should be developed to prevent growth inhibition and increase 1,3-butadiene production. The 1,3-butadiene yield from glucose can also be increased by selecting other *cc*MA pathways or further fine tuning of strains. By continued enzyme development to produce unnatural/nonbiological compounds that are currently made from petroleum, we can develop a circular bioeconomy in a sustainable society.

## Methods

**Chemicals and plasmid construction**. We purchased *cc*MA from Sigma-Aldrich (St. Louis, MO, USA), PA from Active Scientific Sdn Bhd (Kuala Lumpur, Malaysia), and 1,3-butadiene from Tokyo Chemical Industry (Tokyo, Japan). Supplementary Data 1 summarizes the strains and plasmids used in this study. We used *E. coli* NovaBlue-competent cells (Novagen, Cambridge, MA, USA) for gene cloning. We performed polymerase chain reaction (PCR) using KOD-Multi & Epi DNA polymerase (Toyobo, Osaka, Japan) or PrimeSTAR Max (Takara, Shiga, Japan), and the primer pairs are listed in Supplementary Data 2. We assembled each gene with the respective plasmid using NEBuilder HiFi DNA Assembly Master Mix (New England Biolabs, Ipswich, MA, USA).

We constructed pET-T7-*An*FDC vectors as follows. Synthetic genes corresponding to *An*FDC optimized for *E. coli* codon usage were obtained from a commercial source (Invitrogen, Carlsbad, CA, USA) (Supplementary Data 3). A fragment of the *An*FDC gene was amplified via PCR using the synthetic gene *An*FDC as the template in conjunction with the primer pair anfdc_T7_fw and anfdc_T7_rw. The plasmid fragment pSAK-Pt was also amplified via PCR using pSAK-Ptrc as the template in conjunction with the primer pair inv_12_trc_fw and inv_12_trc_rv. Each PCR fragment was conjugated to each other, and the obtained plasmid was named pET-T7-*An*FDC. We constructed pCOLA-T7-UbiX as follows. A fragment of the UbiX gene was amplified via PCR using *E. coli* as the template in conjunction with the primer pair anfdc_T7_fw and ubix_pcola_rv. The plasmid fragment pCOLA-T7 was also amplified via PCR using pCOLADuet-1 as a template in conjunction with the primer pair Inv_pcola_ubix_fw and Inv_pcola_ubix_rv. Each PCR fragment was conjugated to each other, and the obtained plasmid was named pCOLA-T7-UbiX. We constructed pET-T7-AnFDC-UbiX as follows. A fragment of the UbiX gene was amplified via PCR using pCOLA-T7-UbiX as a template in conjunction with the primer pair ubix_duet_fw and ubix_duet_rv. The plasmid fragment pET-T7-AnFDC was also amplified via PCR using pET-T7-AnFDC as a template in conjunction with the primer pair Inv_pet_duet_anfdc_fw and Inv_pet_duet_anfdc_rv. Each PCR fragment was conjugated to each other, and the obtained plasmid was named pET-T7-AnFDC-UbiX.

We constructed pZE12-Ptrc as follows. The *trc* promoter was amplified via PCR using pTrcHis B as a template in conjunction with the primer pair trc_fw and trc_rv. The plasmid fragment pZE12 was also amplified via PCR using pZE12MCS as a template in conjunction with the primer pair inv_12_prom_fw and inv_12_prom_rv. Each PCR fragment was conjugated to each other, and the obtained plasmid was named pZE12-Ptrc. We constructed pZC12Am-Ptrc as follows. The colA gene was amplified via PCR using pCOLADuet-1 as a template in conjunction with the primer pair cola_fw and cola_rv. The plasmid fragment pZE12-Pt was also amplified via PCR using pZE12-Ptrc as a template in conjunction with the primer pair inv_12_ori_fw and inv_12_ori_rv. Each PCR fragment was conjugated to each other, and the obtained plasmid was named pZC12Am-Pt. We constructed pZC12Sp-Ptrc as follows. The gene fragment providing spectinomycin resistance was amplified via PCR using pTargetF as a template in conjunction with the primer pair sp_fw and sp_rv. The plasmid fragment pZC12-Pt was also amplified via PCR using pZC12Am-Ptrc as a template

in conjunction with the primer pair inv_12_res_fw and inv_12_res_rv. Each PCR fragment was conjugated to each other, and the obtained plasmid was named pZC12Sp-Ptrc.

We constructed pSAK-aroF^fbr-tktA as follows. A fragment of the tktA gene was amplified via PCR using *E. coli* K-12 MG1655 genomic DNA as a template in conjunction with the primer pair tkta_psti_fw and tkta_psti_rv. The amplified gene fragment was introduced into the *Pst*I site of pS01. The resultant plasmid was named pSAK-aroF^fbr-tktA. We constructed pZC12Sp-Ptrc-aroF^fbr-tktA as follows: a fragment of the aroF-tktA gene was amplified via PCR using pSAK-aroF^fbr-tktA as a template in conjunction with the primer pair fa_trc_fw and fa_trc_rv. The plasmid fragment pZC12Sp-Ptrc was also amplified via PCR using pZC12Sp-Ptrc as a template in conjunction with the primer pair inv_12_trc_fw and inv_12_trc_rv. Each PCR fragment was conjugated to each other, and the obtained plasmid was named pZC12Sp-Ptrc-aroF^fbr-tktA.

We constructed pSAK-ZYC as follows: a fragment of the ZYC gene was amplified via PCR using pZA23-ZYC as a template in conjunction with the primer pair zyc_fw and zyc_rv. The plasmid fragment pSAK was also amplified via PCR using pSAK as a template in conjunction with the primer pair inv_laca_fw and inv_laca_rv. Each PCR fragment was conjugated to each other, and the obtained plasmid was named pSAK-ZYC. We constructed pSAK-ZYC-UbiX as follows. A fragment of the UbiX was amplified via PCR using pET-T7-AnFDC-UbiX as a template in conjunction with the primer pair ubix_smai_fw and ubix_smai_rv. The amplified fragment was introduced into the SmaI site of pSAK-ZYC, and the obtained plasmid was named pSAK-ZYC-UbiX.

We constructed pSAK-Ptrc as follows. A fragment of the *trc* promoter was amplified via PCR using pTrcHis B as a template in conjunction with the primer pair trc_ps_fw and trc_ps_rv. Fragments of the chloramphenicol resistance and SC101 origin genes were amplified via PCR using pZA33luc and pZS4Int-laci as templates with the primer pairs cmr_fw and cmr_rv and sc101_fw and sc101_rv, respectively. Each PCR fragment was conjugated to each other, and the obtained plasmid was named pSAK-Ptrc. We constructed pSAK-Ptrc-ScFDC1 as follows. A fragment of the *Sc*FDC1 gene was amplified via PCR using the synthetic gene *Sc*FDC1 (Supplementary Data 4) as a template in conjunction with the primer pair scfdc_trc_fw and scfdc_trc_rv. The plasmid fragment pSAK-Pt was also amplified via PCR using pSAK-Ptrc as a template in conjunction with the primer pair inv_12_trc_fw and inv_12_trc_rv. Each PCR fragment was conjugated to each other, and the obtained plasmid was named pSAK-Ptrc-ScFDC1. We constructed pET-T7-*Sc*FDC as follows. A fragment of the *Sc*FDC1 gene was amplified via PCR using pSAK-Ptrc-ScFDC1 as a template in conjunction with the primer pair scfdc_T7_fw and scfdc_T7_rv. The plasmid fragment pET-T7 was also amplified via PCR using pET22b(+) as a template in conjunction with the primer pair Inv_pet_T7_scfdc_fw and Inv_pet_T7_scfdc_rv. Each PCR fragment was conjugated to each other, and the obtained plasmid was named pET-T7-*Sc*FDC. We constructed pET-T7-*Sc*FDC-ScFDC F397H:I398Q as follows: a fragment of the *Sc*FDC gene F397H:I398Q was amplified via PCR using pET-T7-ScFDC F397H:I398Q as a template in conjunction with the primer pair *Sc*FDC F397H:I398Q_duet_fw and *Sc*FDC F397H:I398Q_duet_rv. The plasmid fragment pET-T7-*Sc*FDC was also amplified via PCR using pET-T7-*Sc*FDC as a template in conjunction with the primer pairs Inv_pet_duet_ScFDC F397H:I398Q_fw and Inv_pet_duet_ScFDC F397H:I398Q_rv. Each PCR fragment was conjugated to each other, and the obtained plasmid was named pET-T7-*Sc*FDC-ScFDC F397H:I398Q.

We constructed pET-T7-*Sc*FDC F397H:I398Q-His_tag as follows: a fragment of the *Sc*FDC F397H:I398Q was amplified via PCR using pET-T7-*Sc*FDC F397H:I398Q as a template in conjunction with the primer pair *Sc*FDC His-tag_fw and *Sc*FDC His-tag_rv. The fragment was conjugated and the obtained plasmid was named pET-T7-*Sc*FDC F397H:I398Q-His_tag.

We constructed pTF-N20Δ*ptsHI*::$P_{A1lacO-1}$_*glk_galP* as follows. We amplified *ptsHI*-deficient gene fragments via PCR using *E. coli* K-12 MG1655 gDNA as a template in conjunction with the primer pairs ptshi_l300_fw and ptshi_l300_rv and ptshi_r300_fw and ptshi_r300_rv. We also amplified a $P_{A1lacO-1}$_*glk_galP* cassette via PCR using pCFTdeltain-GG as a template in conjunction with the primer pair lac_gk_gp_fw and lac_gk_gp_rv[21]. Moreover, we amplified the plasmid fragment pTF via PCR using pTargetF as a template in conjunction with the primer pair inv_ptf_fw and inv_ptf_rv. We assembled the three PCR fragments and named the obtained plasmid pTF-Δ*ptsHI*::$P_{A1lacO-1}$_*glk_galP*. To introduce the N20 sequence into pTF-Δ*ptsHI*::$P_{A1lacO-1}$_*glk_galP*, we amplified the plasmid fragment pTF-N20Δ*ptsHI*::$P_{A1lacO-1}$_*glk_galP* via PCR using pTF-Δ*ptsHI*::$P_{A1lacO-1}$_*glk_galP* as a template in conjunction with the primer pair inv_n20_ptshi_fw and inv_n20_ptshi_rw. The resulting fragment self-assembled, and we named the obtained plasmid pTF-N20Δ*ptsHI*::$P_{A1lacO-1}$_*glk_galP*. We constructed pTF-N20Δ*pheA*Δ*tyrA* as follows. We amplified *pheA*- and *tyrA*-deficient gene fragments via PCR using *E. coli* K-12 MG1655 gDNA as a template in conjunction with the primer pairs phea_l300_fw and phea_l300_fw and tyra_r300_fw and tyra_r300_rv, respectively. We also amplified the plasmid fragment pTF via PCR using pTargetF as a template in conjunction with the primer pair inv_ptf_fw and inv_ptf_rv. We assembled the PCR fragments and named the obtained plasmid pTF-Δ*pheA*Δ*tyrA*. To introduce the N20 sequence into pTF-Δ*pheA*Δ*tyrA*, we amplified the plasmid fragment pTF-Δ*pheA*Δ*tyrA* via PCR using pTF-Δ*pheA*Δ*tyrA* as a template in conjunction with the primer pair inv_n20_phea_tyra_fw and inv_n20_phea_tyra_rv. The resultant fragment self-assembled, and we named the obtained plasmid pTF-N20Δ*pheA*Δ*tyrA*.

**Chromosomal gene inactivation.** Using the clustered, regularly interspaced, short palindromic repeat (CRISPR)/CRISPR-associated 9 (Cas9) two-plasmid system together with pTargetF and pCas, we inactivated the tandem gene set of *pheA* and *tyrA* in the chromosome and replaced the tandem gene set of *ptsH* and *ptsI* with the P_{A1lacO-1}_glk_galP cassette[18]. To inactivate *ptsH* and *ptsI* and *pheA* and *tyrA*, we used pTF-N20Δ*ptsHI*::P_{A1lacO-1}_glk_galP and pTF-N20Δ*pheA*Δ*tyrA*, respectively. We named these strains CFB1 and CFB2, respectively.

**Construction of the *cc*MA-FDC-binding model, docking simulation, and in silico mutagenesis.** To obtain a model of *cc*MA-bound *An*FDC, we replaced the binding substrate, α-methyl cinnamic acid, of the substrate-bound crystallographic structure of *An*FDC (PDB:4ZA7) with *cc*MA via the 3D builder tool of MOE 2019.0101. The two models were then protonated using the protonate3D tool of MOE at a pH of 7 and a temperature of 300 K and then optimized by energy minimization using the AMBER10:extended Hückel theory (EHT) force field (gradient = 0.01 RMS kcal mol$^{-1}$ A$^{-2}$). For docking simulation, the force field of AMBER10:EHT and the implicit solvation model of the reaction field (R-field 1:80; cutoff[8,10]) were selected. The docking simulations were carried out with the general dock tool of MOE, and the settings were as follows: site, ligand atoms; ligand, *cc*MA; placement, triangle Matcher method and London ΔG scoring; and refinement, induced fit and GBVI/WSA ΔG scoring. The best scored configurations were selected for further analysis.

The residue scan tool of MOE was used for in silico mutation analysis of *cc*MA-bound *An*FDC, and the settings were as follows: residues, L185, I187, M283, T323, I327, A331, Y394, T395, F397, and L439; mutations, arginine, lysine, histidine, aspartic acid, glutamic acid, serine, threonine, asparagine, glutamine, tyrosine, and cysteine; site limit, 1; and affinity atoms, *cc*MA. The effect of the mutation on the binding free energy (ΔG$_{bind}$) between *cc*MA and the *An*FDC mutants was calculated; the relative binding free affinity changes (Δaffinity) between the *An*FDC mutant (ΔG$_{mutant}$) *An*FDC mutant (ΔG$_{WT}$) were obtained from MOE. The models of multiple *An*FDC mutants were generated with the protein builder tool of MOE, and the models of *cc*MA-bound *An*FDC mutants were constructed as mentioned above.

To obtain a model of *cc*MA-bound *Sc*FDC F397H:I398Q, we first superimposed the generated model of *cc*MA-bound *An*FDC and the crystal structure of *Sc*FDC (PDB:4ZAC) and removed *An*FDC and *An*FDC-binding prFMN. The *cc*MA-bound WT *Sc*FDC model was then constructed as mentioned above. Next, the WT *Sc*FDC model was mutated with the protein builder tool of MOE such that *Sc*FDC F397H: I398Q was generated. The models of *cc*MA-bound *Sc*FDC F397H:I398Q and PA-bound *Sc*FDC F397H:I398Q were constructed as mentioned above.

**Construction and screening of *An*FDC and *Sc*FDC mutants.** After calculating the change in affinity between the *An*FDC mutants and *cc*MA in silico, we selected the top 25 designs on the basis of their affinity. To construct *An*FDC mutants, we performed site-directed mutagenesis via inverse PCR. We amplified pET-T7-*An*FDC plasmids with the specific primer pair required for each mutation. The PCR products were subsequently self-ligated to construct mutated plasmids.

We precultured recombinant *E. coli* cells harboring FDC (WT or mutant *An*FDC) and UbiX in 0.5 mL of Luria–Bertani (LB) medium consisting of 100 μg mL$^{-1}$ ampicillin and 50 μg mL$^{-1}$ kanamycin. After 3 h, 10-μL precultures were added to 1 mL of screening medium consisting of 0.5 mM disodium *cc*MA in a 10-mL HS/GC-MS packed vial. The cultures were then incubated at 37°C in a shaker at 180 rpm. After 18 h, using HS/GC-MS, we analyzed the 1,3-butadiene produced in the gas phase of the vial. The activity of *An*FDC was calculated on the basis of the amount of produced butadiene. The screening medium consisted of (per liter) 20 g of lactose, 12 g of tryptone, 24 g of yeast extract, 12.5 g of K$_2$HPO$_4$, 2.3 g of KH$_2$PO$_4$, antibiotics, and 0.1 mM isopropanol as an internal standard. A disodium *cc*MA stock solution was prepared by mixing *cc*MA and NaOH (two equivalents) together. To further increase the enzymatic activity on *cc*MA, we constructed a combinatorial mutant library, with *An*FDC T395H, T395Q, or T395N used as template enzymes. The enzyme activities of these mutants were measured as mentioned above. *Sc*FDC mutants were also constructed using pET-T7-*Sc*FDC as a template, and these activities were measured as mentioned above.

**Characterization of *Sc*FDC F397H:I398Q.** For analysis of the pH dependence of *Sc*FDC F397H:I398Q activity, we precultured recombinant *E. coli* cells harboring *Sc*FDC F397H:I398Q and UbiX in 0.5 mL of LB medium with antibiotics. After 3 h, we inoculated the cells into 30 mL of screening medium in a 300 mL conical flask and then sealed it with a parafilm. After 18 h, we collected the cells by centrifugation. Afterward, we resuspended the cell pellet in 30 mL of B-PER bacterial cell lysis reagent (Thermo Fisher Scientific, Waltham, MA, USA). After it was shaken for 10 min at 180 rpm, we collected the cell lysate by centrifugation. We added 1 mL of the enzyme assay solution into a 10-mL HS/GC-MS vial and incubated it at 37°C. After 18 h, we analyzed the produced 1,3-butadiene in the gas phase of the vial via HS/GC-MS. The screening medium consisted of 1/10 the volume of cell lysates, 0.5 mM disodium *cc*MA, and 10 mM phosphate buffer. We used 0.1 mM isopropanol as the internal standard.

For analyzing the time course of activity of *Sc*FDC F397H:I398Q, hexa histidine-tagged *Sc*FDC F397H:I398Q and UbiX was also coexpressed in *E. coli.*

*Sc*FDC F397H:I398Q was purified with a His-tag attached to the C-terminal of *Sc*FDC F397H:I398Q using a Ni-NTA column (His-Trap HP column 5 mL, GE Healthcare Bio-Sciences Uppsala, Sweden) in 50 mM phosphate buffer, 50 mM KCl, pH 7, with wash and elution buffers supplemented with 10 and 250 mM imidazole, respectively. Finally, purified His-tagged *Sc*FDC F397H:I398Q was desalted into 20 mM phosphate buffer containing 50 mM KCl (pH 6.0) on PD 10 Sepharose columns (GE Healthcare Bio-Sciences Uppsala, Sweden).

0.1 vvm (volume of gas per volume of liquid per minute) of nitrogen was bubbled through the 100 mM potassium phosphate buffer consisting of 100 mM KCl (pH 6.0) for 30 min. Then, the enzyme assay solution (1 μM *Sc*FDC F397H: I398Q, 50 mM KCL and 50 mM potassium phosphate buffer (pH 6.0)) was prepared. The solutions were incubated 0, 10, 20, 30, 45, 60, 120, 180 min under aerobic or oxygen-depleted conditions. After incubation, 8 mL of the enzyme assay solution was transferred into a 10-mL HS/GC-MS vial, and 8 μL of 500 mM disodium *cc*MA stock was added. The final concentration of ccMA was 0.5 mM. After packing and incubation for 18 h at 37°C, we analyzed the produced 1,3-butadiene in the gas phase of the vial via HS/GC-MS. The activity of *Sc*FDC F397H:I398Q was calculated on the basis of the amount of produced butadiene.

To analyze substrate specificity, *ct*MA was prepared from *cc*MA[62]. One milliliter of ultrapure water was added to 3.55 mg of *cc*MA, which was then heated at 80°C for 30 min at 2,000 rpm. After heating, 50 μL of 1 M NaOH and ultrapure water were added to prepare a 20 mM disodium *ct*MA stock solution. A 20 mM disodium *cc*MA stock solution and a 20 mM disodium *tt*MA stock solution were prepared by mixing NaOH (two equivalents). Sodium (*Z*)-PA (20 mM) was prepared by mixing one equivalent of NaOH. The enzymatic activities of WT *Sc*FDC and *Sc*FDC F397H: I398Q against each substrate were measured as mentioned above.

For co-expression of WT *Sc*FDC and *Sc*FDC F397H:I398Q in *E. coli*, recombinant *E. coli* cells harboring pET-T7-*Sc*FDC_WT-*Sc*FDC_F397H:I398Q and pCOLA-T7-UbiX were cultured, and butadiene production was measured as described above.

All of the experiments with FDC were conducted under light-excluded conditions to prevent the inactivation of FDC by the light[27].

**Culture conditions.** For *cc*MA and 1,3-butadiene production in 5 mL test tube-scale cultures, we used a producing medium consisting of (per liter) 40 g of glucose, 12 g of tryptone, 24 g of yeast extract, 12.5 g of K$_2$HPO$_4$, 2.3 g of KH$_2$PO$_4$, 10 mg of antibiotics, 100 mg of L-phenylalanine, 40 mg of L-tyrosine, and 40 mg of L-tryptophan. We added 0.2 mM isopropyl IPTG to induce protein expression and 10 mM sodium pyruvate to the medium to encourage bacterial growth in the initial phase. We added ampicillin, spectinomycin, and/or chloramphenicol to the medium up to final concentrations of 100, 50, and 15 μg mL$^{-1}$, respectively. We seeded each preculture into 5 mL of this medium in a test tube, and we incubated the test tube-scale cultures at 37°C in a shaker at 180 rpm.

For 1,3-butadiene production, the resultant *E. coli* strains were cultured in a test tube for 24 h, transferred into a 10-mL of HS/GC-MS packed vial and cultured for an additional 24 h. For 1,3-butadiene production under oxygen-depleted conditions, we added 1 mL of the seeded medium to a 10-mL HS/GC-MS packed vial, and the strains were incubated at 37°C in a shaker at 180 rpm. After culture, we analyzed the 1,3-butadiene produced in the gas phase of the vial HS/GC-MS.

We seeded each preculture into 100, 200, and 400 mL of the producing medium in a 2-L medium bottle. The cap had a tube that allowed sampling such that the media and gas phases could be collected with a syringe without opening the bottle. We incubated the oxygen-depleted cultures at 37°C in a shaker at 180 rpm. After measuring the pH, we added a 10% ammonium solution to adjust the pH of the medium to 6.2, if needed. We used M9T medium for [$^{13}C_4$] 1,3-butadiene production in 1-mL HS/GC-MS vial-scale cultures with packing. M9T minimal medium consists of (per liter) 20 g of glucose, 12 g of tryptone, 0.5 g of NaCl, 17.1 g of Na$_2$HPO$_4$·12H$_2$O, 3 g of KH$_2$PO$_4$, 2 g of NH$_4$Cl, 246 mg of MgSO$_4$·7H$_2$O, 14.7 mg of CaCl$_2$·2H$_2$O, 2.78 mg of FeSO$_4$·7H$_2$O, 10 mg of thiamine hydrochloride, 100 mg of L-phenylalanine, 40 mg of L-tyrosine, 40 mg of L-tryptophan, and 0.2 mM IPTG. The percentage of butadiene isotopes was calculated according to the quintic equation. Each peak area distribution was created on the basis of its 1,3-butadiene standard (Supplementary Fig. 9).

$$\begin{pmatrix} 100 & 66.8 & 11.6 & 25.5 & 29 \\ 0 & 100 & 66.8 & 11.6 & 25.5 \\ 0 & 0 & 100 & 66.8 & 11.6 \\ 0 & 0 & 0 & 100 & 66.8 \\ 0 & 0 & 0 & 0 & 100 \end{pmatrix} \begin{pmatrix} a \\ b \\ c \\ d \\ e \end{pmatrix} = \begin{pmatrix} A \\ B \\ C \\ D \\ E \end{pmatrix} \quad (1)$$

*a, b, c, d, e* = relative ratio of each isotope; *m/z* = 54, 55, 56, 57, 58, respectively
*A, B, C, D, E* = each experimental peak area of *m/z* = 54, 55, 56, 57, 58, respectively

$$\text{Percentage of each isotope} = \frac{x}{a+b+c+d+e} \times 100\% \quad (2)$$

*x* = *a, b, c, d,* or *e*.

We performed 1-L jar fermentation with a 400-mL working volume. We added the preculture medium to the medium (400 mL) in a 1-L jar fermenter with an initial OD$_{600}$ of 0.05. The flow rate of air for fermentation was 100 mL min$^{-1}$. To maintain the pH at 6.0 or 7.0 during culture, we added aqueous ammonia to the

medium. We maintained the DO at 0.05–0.5 ppm by automatically controlling the agitation speed from 200 to 800 rpm. The medium for jar fermentation consisted of (per liter) 12 g of tryptone, 24 g of yeast extract, 12.5 g of $K_2HPO_4$, 2.3 g of $KH_2PO_4$, antibiotics, 100 mg of L-phenylalanine, 40 mg of L-tyrosine, 40 mg of L-tryptophan, 0.2 mM isopropyl β- d-1-thiogalactopyranoside (IPTG) and 10 mM sodium pyruvate. The initial glucose concentration was 80 g $L^{-1}$ for batch fermentation. However, for DO-stat fed-batch fermentation, the initial glucose concentration was 50 g $L^{-1}$. The feeding solution was added to the culture medium automatically with a pump when the DO was >0.25 ppm under microaerobic culture conditions (DO of 0~0.05 ppm, after 18 h), and feeding was stopped when the DO was <0.25 ppm. The feeding solution consisted of (per liter) 500 g $L^{-1}$ glucose, 60 g $L^{-1}$ tryptone, and 120 g $L^{-1}$ yeast extract and was added up to 32 mL. We collected the exhaust gas of the culture in HS/GC-MS vials.

**Analytical methods.** We monitored cell growth by measuring the $OD_{600}$ with an UVmini-1240 spectrophotometer (Shimadzu, Kyoto, Japan). We also measured the concentration of glucose in the culture supernatant via a glucose CII test (Wako, Kyoto, Japan) according to the manufacturer's instructions.

We determined the concentrations of *cc*MA in the culture supernatants, which we separated from the culture broth by centrifugation at 15,000 × *g* for 10 min. To accomplish this, we used an organic acid analysis system (Shimadzu) comprising a high-performance liquid chromatography (HPLC) instrument equipped with two Shim-pack SPR-H columns, the first operating at 48℃ and the second operating at 25℃, with a flow rate of 0.8 mL $min^{-1}$. We used a CDD-10A detector, 5 mM p-toluenesulfonic acid (PTSA) as the mobile phase, and 20 mM bis-Tris consisting of 5 mM PTSA. We mixed 100 μM ethylenediaminetetraacetic acid immediately before detection to enhance the sensitivity, and 1 mM pimelic acid was used as an internal standard.

We performed HS/GC-MS via a GC-MS QP2010 Ultra instrument (Shimadzu) equipped with a DB-624 (60 m × 0.32 mm × 1.8 μm) capillary column (Agilent Technologies, Santa Clara, CA, USA) and a HS-20 headspace sampler. We used helium as the carrier gas, and the flow rate was maintained at 75.9 mL $min^{-1}$. The injection volume was 1 μL, with a split ratio of 1:20. We incubated the sample at 40℃ for 15 min. The oven temperature was initially 40℃ for 3 min, after which it was increased to 220℃ at 60℃ $min^{-1}$, was maintained for 2 min, and then was decreased to 40℃ at 80℃ $min^{-1}$, at which point it was maintained for 1 min. The total run time was 11 min. The other settings were as follows: interface temperature, 250℃; ion source temperature, 200℃; and electron impact ionization, 70 eV. We calculated the 1,3-butadiene yield from the concentration of 1,3-butadiene determined in the gas phase of the packed vial. For jar fermentation, we estimated the 1,3-butadiene yield from the concentration of 1,3-butadiene in the exhaust culture gases at each sampling point.

We performed GC-MS using a GC-MS QP2010 Ultra instrument (Shimadzu) equipped with a Pure-WAX (32 m × 0.25 mm × 0.25 μm) column (GL Sciences, Japan). We used helium as the carrier gas, and the flow rate was maintained at 2.12 mL $min^{-1}$. The injection volume was 1 μL, with a split ratio of 1:5. The oven temperature was initially 120℃ for 1 min, after which it was increased to 250℃ at 30℃ $min^{-1}$ at which point it was maintained for 4 min. The total run time was 10 min. The other settings were as follows: interface temperature, 250℃; ion source temperature, 200℃; and electron impact ionization, 70 eV. We determined the PA concentration in the culture supernatant, which we separated from the culture broth by centrifugation at 15,000 × *g* for 15 min. We added 30 μL of 1 M HCl to 300 μL of the culture supernatant. Afterward, the mixture and 300 μL of ethyl acetate were mixed together in a glass vial. After vortexing the mixture for 1 min and centrifuging it at 11,000 × *g* for 1 min, we analyzed the upper phase (Supplementary Fig. 12). We used 0.1 mM hexanoic acid as an internal standard.

**Reporting Summary**. Further information on research design is available in the Nature Research Reporting Summary linked to this article.

## Data availability
Data supporting the findings of this work are available within the paper and its Supplementary Information files. A reporting summary for this Article is available as a Supplementary Information file. The data that support the findings of this study are available from the corresponding author upon reasonable request. Source data are provided with this paper.

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

## Acknowledgements

This work was supported by the FY2017 Incentive Research Projects. We thank Yokohama Rubber Co., Ltd. (Kanagawa, Japan), and the ZEON Corporation (Tokyo, Japan) for financial support.

## Author contributions

T.S. conceived and initiated the research. Y.M. designed and performed the experiments and wrote the paper. S.N. developed the strains, constructed the plasmids, and wrote the relevant methods. T.S. and A.K. supervised the research. All the authors have read and approved the manuscript.

## Competing interests

RIKEN has filed a patent application related to 1,3-butadiene biosynthesis on behalf of Y.M. and T.S. The patent application number is JP 2019-532643. This study was funded by Yokohama Rubber Co., Ltd. (Kanagawa, Japan) and the ZEON Corporation (Tokyo, Japan). The other authors declare no competing interest.
