## [Peer Review File · Nature Communications]

REVIEWER COMMENTS

Reviewer #1 (Remarks to the Author):

The paper demonstrates that the UbiD enzyme Fdc1 can be evolved to accept di-acid substrates, and use a variant evolved for production of pentadienoic acid from cis, cis-muconic acid to support a pathway that produces butadiene from glucose in *E. coli*. Following optimisation of a range of variables, the authors report a max of 2.1 g per L of butadiene produced.

This approach can contribute to a more sustainable approach to hydrocarbon production, but the paper suffers from a range of flaws in my opinion, both scientific and presentational.

In scientific terms, the authors do not mention once that one of the enzymes used in the cis, cis muconic acid pathway is also a UbiD prFMN dependent enzyme: AroY. This is a distant relative from Fdc1 that is shown to be oxygen sensitive in vitro (ref in their manuscript 29). In contrast, not one of the many studies on Fdc1 have reported any detrimental effects of oxygen on activity (only light appears to inactivate the enzyme). Therefore, I do not agree with Fig 1a and their interpretation of the apparent need for semi-anaerobic conditions for butadiene production as due to oxygen mediated inactivation of Fdc1. The activity of AroY will also be dependent on the expression of UbiX.

The authors could easily determine whether the Fdc1 enzymes used (evolved or WT) are oxygen sensitive by determining some kinetic parameters for these enzymes in vitro, a wealth of protocols is available and active holo-Fdc1 is relatively easy to prepare from *E. coli*. Such an analysis would inform the reader whether the evolved variants are likely to be rate limiting under in vivo conditions, and whether cofactor incorporation/stability etc might have been affected by mutagenesis. The authors are overly reliant on computational studies to interpret their results. In this regard, the authors could also then demonstrate whether the enzymes truly work on the cis, cis isomer, or whether the trans/cis or trans/trans isomers are preferred (as occurs for the WT enzyme).

The computational studies and docking are not well documented. Overlays of the docked models with corresponding PBD coordinates used to derive the models should be shown. In a majority of cases the figures indicate an unusual conformation of the prFMN cofactor, where the C4a appears unusually out of the prFMN plane (?). While this might not drastically influence the docking etc, it clearly seems an artifact to me and should be addressed if the computational studies are going to be used for quantitative measurement of protein stability/docking energies etc.

Finally, it seems odd the evolved variant is used to support both decarboxylative steps, given the WT enzyme is likely work with the PA intermediate (for which the evolved variant would have not benefit?).

In presentational terms, the manuscript is overly long in description of some aspects, while others are not clearly shown. The language used needs to be significantly improved both in terms of spelling, grammar and clarity. I have added a range of detailed comments below, but these are by no means the only issues:

labelling of various mutants (ie A1-A25 etc) needs to be improved, presently very un-intuitive and therefore difficult to follow.

P4 66, unclear: "There are three different pathways for 1,3-butadiene production using 66 crotonyl-CoA, erythrose-4-phosphate, and malonyl-CoA, with the biosynthesis of a precursor,19 but no one has yet achieved 1,3-butadiene 67 biosynthesis from glucose as a renewable carbon source."

P5 80 : confusing to state: "However, the decarboxylation mechanism is unclear." When this is clearly

not the case, as later on the text actually confirms

P6 decarboxylate hexadienoic acid, whose structure is very similar to ccMA.

Structure is trans-trans though, so not very similar to ccMA.

P7 104 "keeping a hydrogen bond between the carboxylate. group of natural substrates and FDC." unclear??

Fig 1, DMAP incorrect structure (phosphate oxygen missing), AroY is also prFMN dependent, O₂-inactivated Fdc?? Typo in legend prFMN prenylIFMN

Fig 2a, typo: binig R173

Suppl Fig1, C4a issues??? Overlay with Fig2a needed for clarity

Fig 2b, the X-axis should show the actual mutations, not A1-A25, Y-axis relative activity to what?

Fig 2c, same C4a issue? Overlay with Fig 2a needed

P10 164 "In addition, multiple mutations led to a decrease in the binding affinity between AnFDC and prFMN or changed the position of prFMN in the substrate-binding site, decreasing decarboxylation activity" Exp data??

P12 that a proton form of pH is required ??

P12 with twice the decarboxylation reaction??

Fig 3, a) axis label isn't clear b) C4a issue again, c) japanese/label present, also need to test other substrate

211 The highest yield of ccMA was 0.134 mol mol⁻¹ (explain mol mol⁻¹ of what)??

Fig4 legend labels unclear? Explain more clearly CFB01/CFB11/CFB21
CFB22 UbiX has influence? Line 217, affect on AroY?

Suppl Fig 3b C4a?

Suppl Fig 4, japanese label present

231 a higher oxygen exposure leads to complete inactivation? See above: AroY?? not likely to be Fdc1, no precedent

337 The computer-simulated docking models of these variants and ccMA show displacement of the position of ccMA from its correct position in the substrate-binding site meant for the decarboxylation reaction. Data not shown?

344 The ratio of prFMN species (which ones?) that AnFDC contains is different compared with ScFDC, and inactive prFMN decreases whole-cell activity of 1,3-butadiene production. Data??

347 FDC enzymatic activity requires oxygen to form an active prFMN form; however, more oxygen exposure leads to irreversible oxidative maturation to form an inactive form.28,29??? This is incorrect, 28 and 29 refer to UbiD and AroY respectively, NOT Fdc1

382 In addition, once ccMA and PA are released to the extracellular medium, a proton forms in the medium ???

402 because of the antinomy that prFMN-containing FMN requires oxygen to activate prFMN only once, while more oxygen exposure leads to irreversible inactivation of activated prFMN.28,29 (see above, incorrect assumption here that Fdc1 is oxygen sensitive, it is not).

573 typo: hoboing FDC (AnFDC or ScFDC)

576 typos :10 mL HS/GC-MS bial and packed with a bial cap. We incubated bial-scale

581 typo:E. coli cells hoboing FDC and UbiX

632 typo: gas of the cultivation to the HS/GC-MS bial and packed it with water (?)

The fed batch is not clearly described, what is added and to what conc. Why are butadiene levels "estimated" (see Fig 7/8)

Reviewer #2 (Remarks to the Author):

This study presented the results about designing and modifying FDC, to make E. coli strain producing 1,3-butadiene by decarboxylation of the produced ccMA through the engineered FDC. Previously, 1,3-butadiene have not been biocynthesized directly from glucose in E. coli. FDC, which decarboxylates hexadienoic acid, was assumed to recognize ccM. Therefore, FDC derived from *Aspergillus niger* was designed by computer simulation to improve the FDC-ccMA affinity. Then, the effective amino acid substitutions were applied to FDC from *Saccharomyces cerevisiae*, which showed higher activity than *A. niger* one. Additionally, improved yield of butadiene was achieved by adjusting the dissolved oxygen level and pH of the bioreactor. It is worth considering publication in this journal after considering the following points

Points

#1. It is assumed Y394 and T395 of FDC makes this enzyme efficiently capture ccMA by forming hydrogen bond with carboxylate (opp) group of ccMA. The same enzyme seems to be used to convert pentadienoic acid (PA) to 1,3-butadiene. How would the engineered FDC has affinity with PA when carboxylate (opp) group disappear?

#2. According to Fig. 2 and 3 and the explanation, additional mutations to the A18, A19, and A20, did not produce better enzymes, while those to the S18, S19, and S20 did. What were the activities of wild-type scFDC and AnFDC? Additionally, why did similar mutations in ScFDC mutations show improved performance, but not in AnFDC mutations? How did the additional mutations affect the thermostability of the enzymes?

#. In line 185 on page 11, the descriptions of b and c in figure 3 have been swapped.

#3. What is "metabolic strain" in line 207? Also, brief explanation about CFB11 and CFB21 are needed.

#4. With CFB222, ccMA yield decreased (from 2.59 g/L to 1.27 g/L) but PA yield increased (1.07 g/L) with a small amount of butadiene (44 mg/L). S1905 efficiently decarboxylate ccMA to PA, but does not decarboxylate PA to butadiene. As mentioned above, does this phenomenon due to low affinity of

S1905 to PA?

#5. In line 248 on page 15, "the production rate slowed slower" should be re-written.

#6. In the discussion, please minimize the explanation about the Results, but add more about future direction of the enzyme or process. For example, what are the Gibbs energy of ccMA to PA and PA to butadiene? Butadiene is gas state and how would it affect the equilibrium of the reactions mediated by S1905?

#7. Multiple boxes with Japanese characters added to the figures should be removed.

Reviewer #3 (Remarks to the Author):

This study includes 1) construction of tailored FDC mutants suitable for butadiene production by two-step decarboxylation of ccMA based on in silico simulation, 2) construction of E. coli cells harboring artificial butadiene production pathway, 3) elucidation of important factors for butadiene production by genetically-modified E. coli. Finally, they successfully produced 2.1 g/l of butadiene from glucose in one-batch cultivation process. Especially, results of 1) and 3) are excellent and they are valuable for publication in this journal. I do not have scientific criticism, but there are some points to be considered.

Specific comments:

1. L. 348-351: Is anaerobic condition suitable for ccMA production?
2. Butadiene production at pH6.0 was better than at pH7.0. The authors discussed about effect of pH on intracellular diffusion of substrates into cells at different pHs. But, they also demonstrated that the optimum pH for S1905 is pH6 and at pH7, the activity of S1905 much decreased. Does this pH dependence of S1905 activity contribute to pH dependence of butadiene production?
3. Conditions of DO-stat fed batch cultivation (L. 629-631): Did the author really add 500 g of glucose solution (concentration of glucose?), 60 g of tryptone, and 120 g of yeast extract into 400 ml of medium? What is the unit of DO (>0.25)?

Responses to comments by Reviewer #1:

Thank you for your thoughtful comments that have helped us to improve our manuscript considerably. We revised our manuscript according your suggestion as follows.

Comment 1-1:

In scientific terms, the authors do not mention once that one of the enzymes used in the *cis*, *cis* muconic acid pathway is also a UbiD prFMN dependent enzyme: AroY. This is a distant relative from Fdc1 that is shown to be oxygen sensitive in vitro (ref in their manuscript 29). In contrast, not one of the many studies on Fdc1 have reported any detrimental effects of oxygen on activity (only light appears to inactive the enzyme). Therefore, I do not agree with Fig 1a and their interpretation of the apparent need for semi-anaerobic conditions for butadiene production as due to oxygen mediated inactivation of Fdc1. The activity of AroY will also be dependent on the expression of UbiX.

Response 1-1:

Thank you for your advice.

As you stated, AroY is an enzyme that associates with prFMN as a cofactor. Thus AroY activity was assumed to be reduced under aerobic conditions. However, in aerobic experiments involving test tubes and jar fermenters, the production of *cis*,*cis*-muconic acid (ccMA) was observed (**Fig. 4c**, and **Supplementary Fig. 11b**). ccMA is produced from 3-dehydroshikimate by AroZ, AroY and CatA in our pathway; the ccMA production therefore shows that AroY had the enzymatic activity under the culture conditions employed in this study.

Additionally, as you mentioned, we assumed that AroY activity would be affected by UbiX expression. However, as a result of the ccMA-producing strain used in this study, little difference in ccMA yield was observed with or without UbiX expression (**Fig. 4c** and **Supplementary Fig. 5b**). Therefore, we propose that AroY had sufficient activity to produce ccMA because of the expression of endogenous UbiX.

As you pointed out, we did not show the results regarding oxygen sensitivity of Fdc1. We thus conducted an additional experiment on the oxygen sensitivity of Fdc1.

After the expression of UbiX and ScFDC F397H:I398Q (the best Fdc1 mutant in this study) under oxygen-depleted conditions for 18 h, aeration was started, and the cells were harvested at intervals. With these cells, the whole-cell activity after aeration was measured. As a result, even in the absence of aeration, the whole cells presented activity of butadiene production, and it was assumed that the FDC was activated by oxygen in the air phase of HS/GC-MS vial during the enzyme assay. However, the butadiene production decreased as aeration time increased, and the yield decreased to zero after 180 min. These results indicate that FDC, which is first activated by oxygen, loses its activity during further oxygen exposure. We suggest that this occurs because catalytically active prFMN is oxidized and inactivated, as reported in a study on AroY, resulting in the loss of its decarboxylation function.

Therefore, it is necessary to switch the culture conditions from aerobic to microaerobic conditions for butadiene production to prevent inactivation of Fdc1 by oxygen.

We added an explanatory section discussing AroY as an enzyme that associates with prFMN. We also added an explanatory section discussing the oxygen sensitivity of Fdc1, as well as experimental data.

While AroY, which uses the same coenzyme that prFMN uses, requires exposure to oxygen to induce activity, it is known that overexposure to oxygen causes loss of enzyme activity²⁹. Therefore, we also investigated the effects of oxygen on ScFDC F397H:I398Q. After culturing E. coli coexpressing ScFDC F397H:I398Q and UbiX under oxygen-depleted conditions for 18 h, the culture was aerated. The cells were collected at intervals after aeration started, and whole-cell activity was measured, with ccMA used as a substrate. The relationship between aeration time and the enzymatic activity of FDC is shown in Fig. 3d. Whole cells at 0 min after aeration started presented decarboxylation activity; this activity was defined as 100% at 0 min after the aeration started. The ScFDC F397H:I398Q activity has a half-life of 30 min under aeration conditions but decreased to zero after 180 min. These results indicated that the activity of FDC was lost due to continued oxygen exposure. (Please, see page 10, line 161-page11, line 171)

Fig. 3: Design of ScFDC for 1,3-butadiene production.

d Decay of oxidized ScFDC F397H:I398Q under aeration conditions. The activity was defined as 100% at 0 min after the aeration started. The data are presented as the means \pm SDs of three independent experiments ($n = 3$). ScFDC, ferulic acid decarboxylase derived from *Saccharomyces cerevisiae* SD, standard deviation.

The enzymatic activity of AroY requires oxygen to form an active prFMN; however, increased oxygen exposure leads to irreversible oxidative maturation of prFMN and the formation of a catalytically inactive prFMN. Additionally, it was recently reported that oxidative maturation of prFMN is required for FDC activity.³⁴ The oxygen sensitivity of ScFDC F397H:I398Q in *E. coli* was measured, and loss of enzymatic activity of FDC due to oxygen exposure was observed. Although whole cells in the absence of aeration produced butadiene from ccMA, it was assumed that the FDC was activated by oxygen in the air phase within the headspace/gas chromatography–mass spectrometry (HS/GC-MS) vial during the enzyme assay. (Please, see page 20, line 326-334)

For analyzing the decay of oxidized ScFDC F397H:I398Q in E. coli, we precultured recombinant E. coli cells harboring ScFDC F397H:I398Q and UbiX in 0.5 mL of LB medium with antibiotics. We added the preculture medium to the expression medium (400 mL) in a 1-L jar fermenter (Biott, Tokyo, Japan) with an initial OD₆₀₀ of 0.05 at 37°C. The expression medium consisted of (per liter) 40 g of glucose, 12 g of tryptone, 24 g of yeast extract, 12.5 g of K₂HPO₄, 2.3 g of KH₂PO₄, and antibiotics. The flow rate of air for fermentation was 400 mL min⁻¹, and the agitation speed was 600 rpm. After 4 h, 400 µL of 1 M β-d-1-thiogalactopyranoside (IPTG) was added, and aeration was stopped to start culture and protein expression under oxygen- conditions. After 14 h, aeration was restarted, and 800 µL of culture was collected at 0, 5, 10, 15, 20, 30, 40, 50, 60, 90, 120, 150, and 180 min after the aeration started. After centrifugation at 11,000 × g for 1 min, the supernatants were removed, and the cell pellets were suspended in 8 mL of 50 mM potassium phosphate buffer consisting of 50 mM potassium chloride (pH 6.0). The solution was subsequently transferred into a 10-mL HS/GC-MS vial, and 8 µL of 500 mM disodium ccMA stock was added. The final concentration of ccMA was 0.5 mM. After packing and incubation for 18 h at 37°C, we analyzed the produced 1,3-butadiene in the gas phase of the vial via HS/GC-MS. The activity of ScFDC F397H:I398Q was calculated on the basis of the amount of produced butadiene. (Please, see page 36, line 593-page 37, line 609)

Comment 1-2:

The authors could easily determine whether the Fdc1 enzymes used (evolved or WT) are oxygen sensitive by determining some kinetic parameters for these enzymes in vitro, a wealth of protocols is available and active holo-Fdc1 is relatively easy to prepare from E coli. Such an analysis would inform the reader whether the evolved variants are likely to be rate limiting under in vivo conditions, and whether cofactor incorporation/stability etc might have been affected by mutagenesis. The authors are overly reliant on computational studies to interpret their results. In this regard, the authors could also then demonstrate whether the enzymes truly work on the cis, cis isomer, or whether the trans/cis or trans/trans isomers are preferred (as occurs for the WT enzyme).

Response 1-2:

As noted above, we did not show the oxygen sensitivity of Fdc1. We therefore added a description of the additional experiment. The results of the experiment indicate that Fdc1 activity decrease during oxygen exposure, much like that which occurs for AroY. We therefore did not calculate the kinetic parameters of Fdc1 due to the concern that the activity decreases during enzyme purification and sample preparation, and accurate measurements of the kinetic parameters would thus be highly difficult.

As you suggested, to analyze the substrate specificity of wild-type (WT) ScFDC and ScFDC F397H:I398Q, an enzymatic reaction was performed involving *cis,trans*-muconic acid (*ctMA*) and *trans,trans*-muconic acid (*ttMA*) as substrates, which are structural isomers of *ccMA* used in this study. As a result, we observed that WT ScFDC recognized both *ccMA* isomers as substrates and converted them to butadiene via two decarboxylation reactions. Although ScFDC F397H:I398Q was designed for *ccMA*, 45.8-fold (*ctMA*) and 12.0-fold (*ttMA*) enhancement of butadiene production by ScFDC F397H:I398Q was observed, compared with that of WT ScFDC.

We added these results in our manuscript as follows:

We investigated the substrate specificity of ScFDC F397H:I398Q using the ccMA isomers ctMA and ttMA. We observed that WT ScFDC recognized both ccMA isomers as substrates and converted them to butadiene via two decarboxylation reactions. Although ScFDC F397H:I398Q was designed for ccMA, 45.8-fold (ctMA) and 12.0-fold (ttMA) enhancement of butadiene production by ScFDC F397H:I398Q was observed, compared with that of WT ScFDC. These findings demonstrated that the mutated residues interact with the carboxyl group_{opp} of ccMA, promoting the capture of the carboxyl group_{opp} of ccMA

isomers, and that ScFDC F397H:I398Q efficiently converted them to 1,3-butadiene. (Please, see page 11, line 172-179)

To analyze substrate specificity, ctMA was prepared from ccMA⁶⁰. One milliliter of ultrapure water was added to 3.55 mg of ccMA, which was then heated at 80°C for 30 min at 2,000 rpm. After heating, 50 µL of 1 M NaOH and ultrapure water were added to prepare a 20 mM disodium ctMA stock solution. A 20 mM disodium ccMA stock solution and a 20 mM disodium ttMA stock solution were prepared by mixing NaOH (2 equivalents). Sodium (Z)-pentadienoic acid (20 mM) was prepared by mixing 1 equivalent of NaOH. The enzymatic activities of WT ScFDC and ScFDC F397H:I398Q against each substrate were measured as mentioned above. (Please, see page 37, line 610-617)

Ref 60

Carraher, J. M., Pfennig, T., Rao, R. G., Shanks, B. H. & Tessonier, J.-P. *cis,cis*-Muconic acid isomerization and catalytic conversion to biobased cyclic-C6-1,4-diacid monomers. *Green Chemistry* 19, 3042–3050 (2017).

Comment 1-3:

The computational studies and docking are not well documented. Overlays of the docked models with corresponding PBD coordinates used to derive the models should be shown. In a majority of cases the figures indicate an unusual conformation of the prFMN cofactor, where the C4a appears unusually out of the prFMN plane (?). While this might not drastically influence the docking etc, it clearly seems an artifact to me and should be addressed if the computational studies are going to be used for quantitative measurement of protein stability/docking energies etc.

Response 1-3:

As you stated, the structure of prFMN was distorted in the models. We therefore corrected these structures. The affinity between Fdc1 and ccMA was used as an index for constructing the enzyme mutants in this study. After the structure of prFMN was corrected, the Δ affinity values between the AnFDC mutants and ccMA were recalculated, but the Δ affinity ranking remained unaltered.

All the figures of AnFDC and ScFDC were replaced by reconstructed models with the corrected structure of prFMN.

As you suggested, we added details concerning the computational studies and docking in the methods. In addition, we changed Fig. 2 to an overlay of 4ZA7, which was used as template to construct the substrate-bound AnFDC model, and changed the models of ccMA-bound AnFDC. Along with these changes, we changed the bound-substrate cinnamic acid to α -methyl cinnamic acid in Fig. 2a and in the main text, as α -methyl cinnamic acid was the cocrystallized substrate with AnFDC in 4ZA7.

Construction of the ccMA-FDC binding model, docking simulation and in silico mutagenesis

To obtain a model of ccMA-bound AnFDC, we replaced the binding substrate, α -methyl cinnamic acid, of the substrate-bound crystallographic structure of AnFDC (PDB:4ZA7) with ccMA via the 3D builder tool of MOE 2019.0101. The two models were then protonated using the protonate3D tool of MOE at a pH of 7 and a temperature of 300 K and then optimized by energy minimization using the AMBER10:extended Hückel theory (EHT) force field (gradient = 0.01 RMS kcal mol⁻¹ Å⁻²). For docking simulation, the force field of AMBER10:EHT and the implicit solvation model of the reaction field (R-field 1:80; cut-off8,10) were selected. The docking simulations were carried out with the general dock tool of MOE, and the settings were as follows: site, ligand atoms; ligand, ccMA; placement,

triangle Matcher method and London ΔG scoring; and refinement, induced fit and GBVI/WSA ΔG scoring. The best scored configurations were selected for further analysis.

The residue scan tool of MOE was used for in silico mutation analysis of ccMA-bound AnFDC, and the settings were as follows: residues, L185, I187, M283, T323, I327, A331, Y394, T395, F397, and L439; mutations, arginine, lysine, histidine, aspartic acid, glutamic acid, serine, threonine, asparagine, glutamine, tyrosine, and cysteine; site limit, 1; and affinity atoms, ccMA. The effect of the mutation on the binding free energy (ΔG_{bind}) between ccMA and the AnFDC mutants was calculated; the relative binding free affinity changes ($\Delta_{affinity}$) between the AnFDC mutant (ΔG_{mutant}) AnFDC mutant (ΔG_{WT}) were obtained from MOE. The models of multiple AnFDC mutants were generated with the protein builder tool of MOE, and the models of ccMA-bound AnFDC mutants were constructed as mentioned above.

To obtain a model of ccMA-bound ScFDC F397H:I398Q, we first superimposed the generated model of ccMA-bound AnFDC and the crystal structure of ScFDC (PDB:4ZAC) and removed AnFDC and AnFDC-binding prFMN. The ccMA-bound WT ScFDC model then was constructed as mentioned above. Next, the WT ScFDC model was mutated with the protein builder tool of MOE such that ScFDC F397H:I398Q was generated. The models of ccMA-bound ScFDC F397H:I398Q and PA-bound ScFDC F397H:I398Q were constructed as mentioned above. (Please, see page 32, line 530-page 34 line 558)

Fig. 2: Design of AnFDC for 1,3-butadiene production.

a Overlay of the active site of AnFDC with bound α -methyl cinnamic acid (dark green) from PDB:4ZA7, a model of ccMA (light green)-bound AnFDC with the lowest energy poses and a schematic design for ccMA. The negative hydrogen network and hydrophobic interactions are shown as dashed lines.

Comment 1-4:

Finally, it seems odd the evolved variant is used to support both decarboxylative steps, given the WT enzyme is likely work with the PA intermediate (for which the evolved variant would have not benefit?).

Response 1-4:

The reaction to produce butadiene from ccMA with Fdc1 involves a two-step reaction; ccMA is converted to PA and then PA is decarboxylated to produce butadiene.

When *the* WT ScFDC activity for PA was defined as 1.00, the relative activity of the ScFDC F397H:I398Q was 1.12. However, the combination of WT ScFDC and ScFDC F397H:I398Q resulted in reduced butadiene production from ccMA (**Supplementary Fig. 4b**). Therefore, only ScFDC F397H:I398Q was used in the pathway to produce butadiene from glucose.

We added the results of the FDC enzyme activity for PA, results of the coexpression of WT ScFDC together with ScFDC F397H:I398Q, and an explanation for why the culture experiments were performed using only the mutants in the main text.

We investigated the substrate specificity of ScFDC F397H:I398Q for PA, an intermediate reaction that occurs when muconic acid undergoes a single decarboxylation. The results showed that the relative activity of ScFDC F397H:I398Q for WT ScFDC was 1.12. A ScFDC F397H:I398Q and PA docking model is shown in Supplementary Fig. 4a. We confirmed that substituted F397H, I398Q, and a terminal alkene group of PA were separated. Furthermore, we investigated butadiene production from ccMA by combining WT ScFDC and ScFDC F397H:I398Q. The activity decreased to 56.3% when WT ScFDC and ScFDC F397H:I398Q were coexpressed compared to that when only ScFDC F397H:I398Q was expressed (Supplementary Fig. 4b). Based on these results, it was shown that ScFDC F397H:I398Q can recognize both ccMA and PA as substrates and can efficiently produce 1,3-butadiene from ccMA by a double decarboxylation reaction; thus, only ScFDC F397H:I398Q was used for the butadiene production pathway to be introduced into E. coli. (Please, see page 11, line 180-page 12, line 191)

ScFDC F397:I398Q activity for PA was largely unchanged compared to that of the WT (1.12-fold increase). The PA molecule is smaller than muconic acid is. Therefore, we

assumed that when PA forms a reactive binding state, it is less likely to interact with the substituted amino acid residue F397H:I398Q and would not affect the affinity for PA. When WT ScFDC and ScFDC F397H:I398Q coexisted, the activity on ccMA decreased (56.3%) compared to that when only ScFDC F397H:I398Q was present. While the activity for PA was virtually the same for WT ScFDC and ScFDC F397H:I398Q, ScFDC F397H:I398Q presented extremely high activity for ccMA. Therefore, we assume that the first stage of the reaction from ccMA to PA decreased when WT ScFDC and ScFDC F397H:I398Q coexisted compared to that when only ScFDC F397H:I398Q was expressed. (Please, see page 21, line 336-346)

Comment 1-5:

In presentational terms, the manuscript is overly long in description of some aspects, while others are not clearly shown. The language used needs to be significantly improved both in terms of spelling, grammar and clarity. I have added a range of detailed comments below, but these are by no means the only issues:

Response 1-5:

Thank you for your advice. We sincerely apologize for any unclear or grammatically inadequate sections in the submitted paper. The manuscript has been edited for language accuracy and clarity by Nature Publishing Group Language Editing, and the figures have been corrected.

Comment 1-6:

labelling of various mutants (ie A1-A25 etc) needs to be improved, presently very un-intuitive and therefore difficult to follow.

Response 1-6:

The enzyme labeling was revised as you suggested, and we corrected the labeling to indicate the actual mutants used in the experiment.

Comment 1-7:

P4 66, unclear: "There are three different pathways for 1,3-butadiene production using 66 crotonyl-CoA, erythrose-4-phosphate, and malonyl-CoA, with the biosynthesis of a precursor,¹⁹ but no one has yet achieved 1,3-butadiene ⁶⁷ biosynthesis from glucose as a renewable carbon source."

Response 1-7:

We revised this sentence as follows:

The existence of three different pathways have been suggested for 1,3-butadiene production from glucose via crotonyl-CoA, erythrose-4-phosphate, and malonyl-CoA,¹⁹ but

the direct 1,3-butadiene biosynthesis from glucose as a renewable carbon source has not been achieved. (Please, see page 4, line 62-65)

Comment 1-8:

P5 80: confusing to state: "However, the decarboxylation mechanism is unclear." When this is clearly not the case, as later on the text actually confirms

Response 1-8:

We removed this statement and revised our manuscript as follows:

Ferulic acid decarboxylase (FDC), a member of the UbiD family enzymes, mediates the decarboxylation of phenylacrylic acid derivatives and converts them to terminal alkenes.²⁴ The novel cofactor prenylated flavin mononucleotide (prFMN) was recently discovered, and it was revealed that FDC-binding prFMN catalyzes the decarboxylation reaction.²⁵ (Please, see page 5, line 74-77)

Comment 1-9:

P6 decarboxylate hexadienoic acid, whose structure is very similar to ccMA.

Structure is trans-trans though, so not very similar to ccMA.

Response 1-9:

As you remarked, we found the structure to be different; thus, we revised our manuscript as follows:

prFMN-binding FDC can recognize not only aromatic compounds but also α,β -unsaturated carboxylic acids to produce terminal alkenes.^{27,28,29,30,31,32,33,34} Therefore, this FDC was selected as a template enzyme for producing butadiene from the α,β -unsaturated dicarboxylic acid ccMA. (Please, see page 5, line 80-83)

Comment 1-10:

P7 104 "keeping a hydrogen bond between the carboxylate. group of natural substrates and FDC." unclear??

Response 1-10:

We revised this sentence as follows:

ccMA decarboxylation reactions were assumed to be facilitated by an amino acid substitution at the FDC substrate-binding site. This substitution would result in the capture of the carboxylate_{opp.} group while those amino acid residues (R173 and E282) involved in the decarboxylation reaction would be conserved. (Please, see page 7, line113-116)

Comment 1-11:

Fig 1, DMAP incorrect structure (phosphate oxygen missing), AroY is also prFMN dependent, O₂- inactivated Fdc??

Response 1-11:

We corrected the structure of DMAP.

As mentioned above (**please see Response 1-1**), we added an explanation about AroY and the oxygen sensitivity of Fdc1 in the main text.

Comment 1-12:

Typo in legend prFMN prenylFFMN

Response 1-12:

We have corrected the misspelled word in legend Fig.1.

Comment 1-13:

Fig 2a, typo: bining R173

Response 1-13:

We have corrected the misspelled word in Fig 2a.

Comment 1-14:

Suppl Fig1, C4a issues??? Overlay with Fig2a needed for clarity

Fixed the structure of prFMN in all model diagrams, including Supplemental Figure 1. In addition, Fig2a was changed to an overlay diagram with the crystal structure data (PDB: 4ZA7) used to construct the protein-substrate binding model.

Response 1-14:

We corrected the structure of prFMN in all the model diagrams, including those in Supplementary Fig. 1. We also changed Fig. 2a to an overlay diagram including the crystalline structure data (PDB: 4ZA7) used to construct the protein-substrate binding model (**please see Response 1-3**).

Comment 1-15:

Fig 2b, the X-axis should show the actual mutations, not A1-A25, Y-axis relative activity to what?

Response 1-15:

The labels of the mutants in the manuscript were changed to the actual mutations that were introduced. The vertical axis of Fig. 2b indicates the relative activity when the activity of WT *AnFDC* was defined as 1.0. The activity was calculated on the basis of the amount of produced butadiene. An explanation concerning the relative activity was added in the Fig 2b legend.

Fig. 2: Design of *AnFDC* for 1,3-butadiene production.

*b Relative decarboxylation activity of 25 *AnFDC* mutants designed for ccMA. The activity of WT *AnFDC* was defined as 1. The data are presented as the means \pm SDs of three independent experiments ($n = 3$).*

Comment 1-16:

Fig 2c, same C4a issue? Overlay with Fig 2a needed

Response 1-16:

We corrected the structure of prFMN. We also changed Fig. 2a to an overlay model with the crystalline structure data (PDB: 4ZA7) used to construct the protein-substrate binding model and ccMA-bound AnFDC model (**please see Response 1-3**).

Comment 1-17:

P10 164 "In addition, multiple mutations led to a decrease in the binding affinity between AnFDC and prFMN or changed the position of prFMN in the substrate-binding site, decreasing decarboxylation activity" Exp data??

Response 1-17:

"As a calculation, affinity with prFMN was reduced, and the position was changed within the substrate binding pocket", which was assumed to be the reason for the reduced activity due to multiple mutations. However, we have no experimental data to confirm this assumption.

Furthermore, as you pointed out, it became apparent that the structure of prFMN used in the simulation was distorted; therefore, this sentence was deleted.

Comment 1-18:

P12 that a proton form of pH is required ??

Response 1-18:

We revised our manuscript as follows:

Considering a pKa of the substituted His residue of ~6.0, we assumed that the protonated form of the imidazole side chain of His is required for efficient binding of ccMA to ScFDC F397H:I398Q. (Please, see page 20, line 323-325)

Comment 1-19:

P12 with twice the decarboxylation reaction??

Response 1-19:

We have revised our manuscript as follows:

Based on these results, it was shown that ScFDC F397H:I398Q can recognize both ccMA and PA as substrates and can efficiently produce 1,3-butadiene from ccMA by a double decarboxylation reaction; thus, only ScFDC F397H:I398Q was used for the butadiene production pathway to be introduced into E. coli. (Please, see page 12, line 188-191)

Comment 1-20:

Fig 3, a) axis label isn't clear b) C4a issue again, c) japanese/label present, also need to test other substrate

Response 1-20:

We changed the label on the horizontal axis to indicate the actual mutation that was introduced, we corrected the structure of prFMN, and we removed the Japanese characters. As mentioned above (**please see Response 1-2**), the substrate specificity was measured via ccMA isomers.

Comment 1-21:

211 The highest yield of ccMA was 0.134 mol mol⁻¹ (explain mol mol⁻¹ of what)??

Response 1-21:

This indicated the ccMA yield relative to glucose. An explanation of the units mol mol⁻¹ was added as follows:

The highest yield of ccMA was 0.134 mol (mol glucose)⁻¹, which occurred from the CFB21 culture (Supplementary Table 3). (Please, see page 13, line 210-211)

Comment 1-22:

Fig4 legend labels unclear? Explain more clearly CFB01/CFB11/CFB21

Response 1-22:

The ccMA-producing strains used in this study include the following:

CFB01, a strain in which the gene cluster required for ccMA production is inserted in *Escherichia coli* C41 (DE3) for protein expression;

CFB11, a strain in which the gene cluster associated with ccMA production is inserted in the *E. coli* C41 (DE3) ptsHI::PA1lacO-1-Glk-GalP strain; and

CFB21, a strain in which the gene cluster associated with ccMA production is inserted in the *E. coli* C41 (DE3) ptsHI::PA1lacO-1-Glk-GalP, $\Delta phe \Delta tyrA$ strain.

We corrected the legend in Fig. 4 and added an explanation of the strains in the main text.

In this ccMA pathway via PCA, ccMA is produced from 3-dehydroshikimic acid (3DHS) by 3-DHS dehydratase (aroZ), protocatechuic acid decarboxylase (aroY), and catechol dioxygenase (catA) in a three-step reaction. We selected aroZ derived from Bacillus thuringiensis, aroY from Klebsiella pneumoniae, and catA from Pseudomonas putida DOT-T1E. For ccMA production, these genes were incorporated into CFB01, CFB11, and CFB21, as shown in Supplementary Table 4; these genes constitute the basis of aromatic derivative-producing E. coli strains (Fig. 4).^{21,23} (Please, see page 12, line 199-page 13, line 205)

Comment 1-23:

CFB22 UbiX has influence? Line 217, affect on AroY?

Response 1-23:

As you mentioned, we assumed that AroY activity would be affected by UbiX expression. However, as a result of the ccMA-producing strain used in this study, little

difference in ccMA yield was observed with or without UbiX expression (**Fig. 4c** and **Supplementary Fig. 5b**). Therefore, we propose that AroY had sufficient activity to produce ccMA due to the expression of endogenous UbiX.

We have added this explanation in the main text.

AroY, which generates CAT from PCA in the ccMA production pathway, is an enzyme that associates with prFMN as a coenzyme; therefore, we assumed that AroY activity would be affected by UbiX expression in the same way as FDC was. However, in vitro culturing of ccMA-producing bacteria did not result in an increase in ccMA production regardless of UbiX expression. This may have occurred because AroY was sufficiently active to produce ccMA due to prFMN produced by endogenous UbiX. Although it is possible that AroY activity is reduced by oxygen, ccMA was produced by AroY and also by CatA in an experiment in which test tubes and jars were used under aerobic conditions, demonstrating that AroY has sufficient activity under these experimental conditions. (Please, see page 22, line 361-369)

Comment 1-24:

Suppl Fig 3b C4a?

Response 1-24:

All the figures of AnFDC and ScFDC were replaced by reconstructed models with the corrected structure of prFMN (**Please see Response 1-3**).

Comment 1-25:

ASuppl Fig 4, japanese label present

Response 1-25:

We removed the Japanese characters.

Comment 1-26:

231 a higher oxygen exposure leads to complete inactivation? See above: AroY?? not likely to be Fdc1, no precedent

Response 1-26:

As mentioned above, we added an explanation of AroY, an explanation of the oxygen sensitivity of Fdc1, and experimental data to the text. **(Please see Response 1-1)**

Comment 1-27:

337 The computer-simulated docking models of these variants and ccMA show displacement of the position of ccMA from its correct position in the substrate-binding site meant for the decarboxylation reaction. Data not shown?

Response 1-27:

The docking simulation with an *AnFDC* mutant, which presented low activity, was conducted, and we confirmed that the ccMA-binding site was displaced.

Figure for revision: Comparison of ccMA conformations in the active site of *AnFDC* mutants. a the model of ccMA (light green)-bound *AnFDC* Y394H:T395Q with the lowest

energy docking poses. **b, c, d, e** and **f** the model of ccMA (purple)-bound *AnFDC* I187H:A331T:Y394N:T395Q with the top 5 of the lowest docking poses.

AnFDC Y394H:T395Q, the best design of *AnFDC* mutants in this study; *AnFDC* I187H:A331T:Y394N:T395Q, one of the *AnFDC* variants with multiple mutations, which presented low activity.

In the best design of *AnFDC* mutant, the double bond of ccMA located in proximity to the C1' and C4a atoms of the prFMN (Figure for revision **a**, light green). In the *AnFDC* mutant with reduced activity, the position of ccMA with the lowest energy docking pose (Figure for revision **b**, purple) was displaced from the correct position for the decarboxylation reaction in the docking simulation. The ccMA with the correct position was the second lowest energy docking pose (Figure for revision **c**), and other displacements were observed (Figure for revision **d, e** and **f**). In other *AnFDC* mutants with reduced activity, these displacements in the docking simulation were also confirmed.

However, we have no experimental data to confirm this displacement. Given that the activity did not increase even for multiple mutations, the above sentence was removed.

Comment 1-28:

344 The ratio of prFMN species (which ones?) that *AnFDC* contains is different compared with *ScFDC*, and inactive prFMN decreases whole-cell activity of 1,3-butadiene production. Data??

Response 1-28:

Compared with the *AnFDC* mutant, the *ScFDC* mutant presented 1.59-fold higher activity (**Supplementary Fig. 3**); however, we have no data to indicate that this is due to differences in the active:inactive prFMN ratio.

This section was revised as follows:

*The difference in decarboxylase activity between *AnFDC* Y394H:T395Q and *ScFDC* F397H:I398Q may depend on the differences in the ratio of the active/inactive prFMN contained.^{25,27,34} (Please, see page 20, line 334-page 21, line 336)*

Comment 1-29:

347 FDC enzymatic activity requires oxygen to form an active prFMN form; however, more oxygen exposure leads to irreversible oxidative maturation to form an inactive form.^{28,29} This is incorrect, 28 and 29 refer to UbiD and AroY respectively, NOT Fdc1

Response 1-29:

As you suggested, we cited articles on UbiD and AroY. Recently, it was reported that the catalytic activity of prFMN-bound ScFDC is activated by oxygen due to the maturation of prFMN.[Ref 34] Thus, we cited this article as a reference and added experimental concerning Fdc1 oxygen sensitivity. In addition, we added an explanation about the activation of ScFDC in the main text.

The enzymatic activity of AroY requires oxygen to form an active prFMN; however, increased oxygen exposure leads to irreversible oxidative maturation of prFMN and the formation of a catalytically inactive prFMN. Additionally, it was recently reported that oxidative maturation of prFMN is required for FDC activity.³⁴ The oxygen sensitivity of ScFDC F397H:I398Q in E. coli was measured, and loss of enzymatic activity of FDC due to oxygen exposure was observed. (Please, see page 20, line 326-331)

Ref 34

Balaikaite, A. et al. Ferulic Acid Decarboxylase Controls Oxidative Maturation of the Prenylated Flavin Mononucleotide Cofactor. *ACS Chem. Biol.* **15**, 2466–2475 (2020)

Comment 1-30:

382 In addition, once ccMA and PA are released to the extracellular medium, a proton forms in the medium ???

Response 1-30:

Considering the pKa value, nearly all ccMA and PA are present in the COO⁻ state in the media. However, these must be in a protonated form to permeate the *E. coli* membrane. Therefore, we assumed that low pH, conditions in which the substrate, once released,

tends to form a protonated form, promotes the uptake of ccMA and PA, leading to increased butadiene yield.

We revised manuscript as follows:

The calculated pKa values are 4.89 for PA and 3.87 and 4.65 for ccMA, indicating that the proportion of PA and ccMA protonated forms increases under low-pH conditions. The protonated forms of ccMA and PA can reenter cells in a medium, which may promote the conversion to butadiene. (Please, see page 23, line 374-377)

Comment 1-31:

402 because of the antinomy that prFMN-containing FMN requires oxygen to activate prFMN only once, while more oxygen exposure leads to irreversible inactivation of activated prFMN.^{28,29} (see above, incorrect assumption here that Fdc1 is oxygen sensitive, it is not).

Response 1-31:

We revised the citation, added experimental data concerning Fdc1 oxygen sensitivity, and added a relevant phrase to the main text (**please see Response 1-1**).

Comment 1-32:

573 typo: hoboing FDC (AnFDC or ScFDC)

Response 1-32:

We corrected the misspelled word.

Comment 1-33:

576 typos :10 mL HS/GC-MS bial and packed with a bial cap. We incubated bial-scale

Response 1-33:

We revised the misspelled word.

Comment 1-34:

581 typo: E. coli cells hoboining FDC and UbiX

Response 1-34:

We corrected the misspelled word.

Comment 1-35:

632 typo: gas of the cultivation to the HS/GC-MS bial and packed it with water (?)

Response 1-35:

Butadiene is a gaseous substance and is present in exhaust gas during jar fermentation. Therefore, exhaust gas was collected in a vial, and the amount of butadiene produced was measured.

We revised our manuscript as follows:

We collected the exhaust gas of the culture in HS/GC-MS vials. (Please, see page 40, line 661-662)

Comment 1-36:

The fed batch is not clearly described, what is added and to what conc. Why are butadiene levels "estimated" (see Fig 7/8)

Response 1-36:

The target product, butadiene, is a gas that is insoluble in the medium, and it is constantly released as an exhaust gas during culturing. Therefore, exhaust gas was collected with a syringe, and the concentration at the time of sampling was integrated to estimate the amount of butadiene production.

We added details of experiments about Fed batching.

We performed 1-L jar fermentation with a 400-mL working volume. We added the preculture medium to the medium (400 mL) in a 1-L jar fermenter with an initial OD_{600} of 0.05. The flow rate of air for fermentation was 100 mL min^{-1} . To maintain the pH at 6.0 or 7.0 during culture, we added aqueous ammonia to the medium. We maintained the DO at 0.05–0.5 ppm by automatically controlling the agitation speed from 200 to 800 rpm. The medium for jar fermentation consisted of (per liter) 12 g of tryptone, 24 g of yeast extract, 12.5 g of K_2HPO_4 , 2.3 g of KH_2PO_4 , antibiotics, 100 mg of L-phenylalanine, 40 mg of L-tyrosine, 40 mg of L-tryptophan, 0.2 mM IPTG and 10 mM sodium pyruvate. The initial glucose concentration was 80 g L^{-1} for batch fermentation. However, for DO-stat fed-batch fermentation, the initial glucose concentration was 50 g L^{-1} . The feeding solution was added to the culture medium automatically with a pump when the DO was $> 0.25 \text{ ppm}$ under microaerobic culture conditions (DO of 0~0.05 ppm, after 18 h), and feeding was stopped when the DO was $< 0.25 \text{ ppm}$. The feeding solution consisted of (per liter) 500 g L^{-1} glucose, 60 g L^{-1} tryptone, and 120 g L^{-1} yeast extract and was added up to 32 mL. We collected the exhaust gas of the culture in HS/GC-MS vials. (Please, see page 39, line 648-page 40, line 662)

Responses to comments by Reviewer #2:

Thank you very much for valuable comments on our manuscript. We have revised our manuscript according to your suggestions.

Comment 2-1:

It is assumed Y394 and T395 of FDC makes this enzyme efficiently capture ccMA by forming hydrogen bond with carboxylate (opp) group of ccMA. The same enzyme seems to be used to convert pentadienoic acid (PA) to 1,3-butadiene. How would the engineered FDC has affinity with PA when carboxylate (opp) group disappear?

Response 2-1:

As you suggested, we assumed that pentadienoic acid (PA) cannot be captured by amino acid residues that should interact with the carboxylate (opp) group of engineered FDC. However, PA is captured by amino acid residues that interact with the carboxylate (react) group (**Supplementary Fig. 4a**), and the reaction occurs simultaneously.

Comment 2-2:

According to Fig. 2 and 3 and the explanation, additional mutations to the A18, A19, and A20, did not produce better enzymes, while those to the S18, S19, and S20 did. What were the activities of wild-type scFDC and AnFDC? Additionally, why did similar mutations in ScFDC mutations show improved performance, but not in AnFDC mutations? How did the additional mutations affect the thermostability of the enzymes?

Response 2-2:

The activities of ScFDC and AnFDC were evaluated according to the amount of 1,3-butadiene produced after 18 h. The activity of the WT ScFDC and WT AnFDC was defined as 1.0, and the activity of each mutant was expressed as the relative activity.

The activity was also increased by introducing additional mutations to A18 (AnFDC T395N), A19 (AnFDC T395Q), and A20 (AnFDC T395H), and the A1905 (AnFDC Y394H:T395Q) mutant with one mutation added to A19 presented the greatest activity among AnFDC mutants (**Supplementary Table 2**). A similar increase in activity was

achieved when well-designed *AnFDC* mutants that showed increased activity were applied to ScFDC (**Fig. 3a**).

The enzyme mutant label was changed to the actual mutation that was introduced to clarify this section.

The computational calculation showed a decrease in thermostability by multiple mutations because of the incorporation of several hydrophilic residues into the hydrophobic cluster region in the substrate-binding site ($\Delta\text{stability} > 3.0 \text{ kcal mol}^{-1}$). However, we have no experimental data showing a decrease in thermostability, and the activity of enzymes with three mutations or more did not increase more than the activity of *AnFDC* Y394H:T395Q (A1905) and ScFDC F397H:I398Q (S1905).

Comment 2-3:

#. In line 185 on page 11, the descriptions of b and c in figure 3 have been swapped.

Response 2-3:

Thank you for your advice. We have corrected the description in Fig. 3.

Comment 2-4:

What is "metabolic strain" in line 207? Also, brief explanation about CFB11 and CFB21 are needed.

Response 2-4:

We have changed the words to "engineered strain" and added a description of the *E. coli* strain used in this study in the main text.

CFB11 and CFB21, the engineered strains designed for ccMA production, continued to grow and produce ccMA after 24 h of culture (Please, see page 13, line 206-208).

In this ccMA pathway via PCA, ccMA is produced from 3-dehydroshikimic acid (3DHS) by 3-DHS dehydratase (aroZ), protocatechuic acid decarboxylase (aroY), and catechol dioxygenase (catA) in a three-step reaction. We selected aroZ derived from Bacillus thuringiensis, aroY from Klebsiella pneumoniae, and catA from Pseudomonas putida DOT-T1E. For ccMA production, these genes were incorporated into CFB01, CFB11, and CFB21, as shown in Supplementary Table 4; these genes constitute the basis of aromatic derivative-producing E. coli strains (Fig. 4).^{21,23} (Please, see page 12, line 199-page 13, line 205)

Comment 2-5:

With CFB222, ccMA yield decreased (from 2.59 g/L to 1.27 g/L) but PA yield increased (1.07 g/L) with a small amount of butadiene (44 mg/L). S1905 efficiently decarboxylate ccMA to PA, but does not decarboxylate PA to butadiene. As mentioned above, does this phenomenon due to low affinity of S1905 to PA?

Response 2-5:

Yes, this is correct. We assumed that ScFDC F397H:I398Q (S1905) had increased affinity for ccMA compared to that of WT ScFDC, but the affinity for PA did not increase. In essence, the activity of WT ScFDC and ScFDC F397H:I398Q for PA did not change substantially (1.12-fold increase).

We have added a description in the main text.

We investigated the substrate specificity of ScFDC F397H:I398Q for PA, an intermediate reaction that occurs when muconic acid undergoes a single decarboxylation. The results showed that the relative activity of ScFDC F397H:I398Q for WT ScFDC was 1.12. A ScFDC F397H:I398Q and PA docking model is shown in Supplementary Fig. 4a. We confirmed that substituted F397H, I398Q, and a terminal alkene group of PA were separated. (Please, see page 11, line 180-184)

ScFDC F397:I398Q activity for PA was largely unchanged compared to that of the WT (1.12-fold increase). The PA molecule is smaller than muconic acid is. Therefore, we assumed that when PA forms a reactive binding state, it is less likely to interact with the substituted amino acid residue F397H:I398Q and would not affect the affinity for PA. (Please, see page 21, line 336-340)

Comment 2-6:

In line 248 on page 15, "the production rate slowed slower" should be re-written.

Response 2-6:

Thank you for your advice. Our revised manuscript has been edited by Nature Publishing Group Language Editing once more, and we revised this sentence as follows:

Culturing (during which the pH was adjusted) was subsequently performed to examine the effects of pH on butadiene production (Supplementary Fig. 7). Butadiene production was higher at a low pH than at a pH of approximately 7.0, which is optimal for E. coli growth. (Please, see page 15, line 245-248)

Comment 2-7:

In the discussion, please minimize the explanation about the Results, but add more about future direction of the enzyme or process. For example, what are the Gibbs energy of ccMA to PA and PA to butadiene? Butadiene is gas state and how would it affect the equilibrium of the reactions mediated by S1905?

Response 2-7:

Thank you for your advice. We minimized the explanation in the discussion section and added a statement about future directions. As you advised, we also added a description of Gibbs free energy as follows:

We constructed a novel artificial metabolic pathway for 1,3-butadiene from glucose by combining a ccMA-producing pathway with the decarboxylation of ccMA by FDC. The butadiene-producing reaction in this study involves a two-step reaction that uses ccMA as an initial substrate, with PA generated from ccMA (first step) and butadiene generated from PA (second step) by decarboxylation reactions. The standard Gibbs free energy of the reaction is $\Delta G^{0'} = -14.2 \text{ kJ mol}^{-1}$ for both reactions, making it thermodynamically.^{54, 55, 56} At room temperature, butadiene is gaseous and insoluble in water; thus, the second step of the reaction is associated with a strong positive value. This reaction is therefore considered

extremely useful as an artificial metabolic pathway for butadiene production. (Please, see page 21, line347-page 22, line 355)

*The intermediate substrates ccMA and PA remained in the medium at the end of fermentation. This means that the activity of the developed ScFDC mutant is still insufficient. Therefore, we hypothesize that butadiene production may be substantially increased when a mutant with increased affinity for PA is used and when that mutant is combined with ScFDC F397H:I398Q, as developed in this study. FDC mutant enzyme activity depends on the pH, of which the optimum is 6.0, and enzymatic activity may be considerably reduced when the *E. coli* intracellular pH is 6.8–7.5. Therefore, we assumed that changing the optimal pH of FDC mutants to 6.8–7.5 would promote FDC activity in *E. coli* and butadiene production. (Please, see page 24, line 387-394)*

Comment 2-8:

Multiple boxes with Japanese characters added to the figures should be removed.

Response 2-8:

We removed Japanese characters in the figures.

Responses to comments by Reviewer #3:

Thank you very much for the numerous insightful comments on our manuscript. We revised the manuscript according to your suggestions as follows.

Comment 3-1:

L. 348-351: Is anaerobic condition suitable for ccMA production?

Response 3-1:

Thank you for your comments. We apologize the incorrect sentence (L. 348-351 before revision). In this pathway, oxygen is required for prFMN activation and ccMA synthesis by *catA*, while oxygen-depleted conditions are needed to maintain FDC activity.

We revised our manuscript as follows:

Therefore, in this pathway for 1,3-butadiene production, aerobic conditions are suitable for the activation of FDC and for ccMA production in the early stage of culture, whereas oxygen-depleted conditions are needed to maintain FDC activity. (Please, see page 14, line 235-page 15, line 238.)

*Oxygen is required for prFMN activation and ccMA synthesis by *catA*, while oxygen-depleted conditions are needed to maintain FDC activity. (Please, see page 22, line 356-357.)*

Comment 3-2:

Butadiene production at pH6.0 was better than at pH7.0. The authors discussed about effect of pH on intracellular diffusion of substrates into cells at different pHs. But, they also demonstrated that the optimum pH for S1905 is pH6 and at pH7, the activity of S1905 much decreased. Does this pH dependence of S1905 activity contribute to pH dependence of butadiene production?

Response 3-2:

We do not assume that there is direct contribution.

The reason is that even if the pH of the medium is adjusted to 6.0, the intracellular pH is assumed to be maintained at approximately 7.0. (Ref 59) Thus, the intracellular activity of ScFDC F397H:I398Q (S1905) may be lower than that at a pH of 6.0. As such, a further increase in butadiene production would be expected through development of an enzyme whose activity does not decrease at a pH of 7.0.

This information was added to the future research outlook section of the discussion.

FDC mutant enzyme activity depends on the pH, of which the optimum is 6.0, and enzymatic activity may be considerably reduced when the E. coli intracellular pH is 6.8–7.5⁵⁹. Therefore, we assumed that changing the optimal pH of FDC mutants to 6.8–7.5 would promote FDC activity in E. coli and butadiene production. (Please, see page 24, line 391-394)

Ref 59

Martinez, K. A. *et al.* Cytoplasmic pH Response to Acid Stress in Individual Cells of Escherichia coli and Bacillus subtilis Observed by Fluorescence Ratio Imaging Microscopy. *Applied and Environmental Microbiology* **78**, 3706–3714 (2012).

Comment 3-3:

Conditions of DO-stat fed batch cultivation (L. 629-631): Did the author really add 500 g of glucose solution (concentration of glucose?), 60 g of tryptone, and 120 g of yeast extract into 400 ml of medium? What is the unit of DO (>0.25)?

Response 3-3:

We apologize. The above phrase is incorrect. The dissolved oxygen (DO) units are parts per million (ppm; mg kg⁻¹). We revised our manuscript as follows:

The feeding solution was added to the culture medium automatically with a pump when the DO was > 0.25 ppm under microaerobic culture conditions (DO of 0~0.05 ppm, after 18 h), and feeding was stopped when the DO was < 0.25 ppm. The feeding solution consisted of (per liter) 500 g L⁻¹ glucose, 60 g L⁻¹ tryptone, and 120 g L⁻¹ yeast extract and was added up to 32 mL. (Please, see page 40, line 657-662)

REVIEWER COMMENTS

Reviewer #1 (Remarks to the Author):

The manuscript is now much improved following the changes made in response to reviewers comments.

I have however one main outstanding concern, which relates to the observation that oxygen levels are key to butadiene production. While I do not contest this interpretation, I do take issue with the interpretation by the authors this is due to Fdc inactivation by oxygen.

I repeat my initial comment this has not been observed during any Fdc in vitro studies reported (be it from *A. niger* or *S. cerevisiae*), of which there are now many from various groups. Light mediated inactivation has been reported to occur in vitro.

The authors now present Fig 3d, as evidence that ScFdc is inactivated by oxygen. It is unclear what this refers to (ie pure protein/in vitro or is it cells containing ScFdc/in vivo)?

I suspect the latter, which could be consequence of a whole range of issues:

- 1) in vivo production of prFMN-OOH which does inhibit Fdc (see ref 34)
- 2) mechanism based inhibition, oxygen somehow inhibiting the ScFdc (mutants or WT) during turnover
- 3) light mediated inhibition (has light been excluded?)

I again must repeat my initial comments that Fdc is easily purified from *E. coli* and has not been found to be particularly difficult to study. Therefore, my original request to delve a bit deeper into the actual apparent oxygen-inactivation through in vitro studies of the evolved variants stands.

As a minimum, the authors should clearly state they have no direct evidence for the process by which oxygen might inactivate the Fdc in vivo activity.

My second main comment is about good reporting of values in the text of the main manuscript, which are frequently (if any!) reported without error, and I suspect almost all with precision that are beyond the accuracy of the measurement. This should be checked throughout and corrected. As an example see: line 186, top of p12: 56.3% ? (this suggest the error is in the range of 0.1%?), top of p13 line 208: 2.17 and 2.52 (this suggest errors of 0.01?) and later 0.134 mol (so +- 0.001 mol?)

Reviewer #2 (Remarks to the Author):

The manuscript was revised properly. I would recommend to accept it for publication.

Reviewer #3 (Remarks to the Author):

The authors have properly revised the MS based on the reviewer's comments.

Responses to comments by Reviewer #1:

Thank you for your thoughtful comments that have helped us to improve our manuscript considerably. We revised our manuscript according your suggestion as follows.

Comment 1-1:

I repeat my initial comment this has not been observed during any Fdc in vitro studies reported (be it from *A niger* or *S. cerevisiae*), of which there are now many from various groups. Light mediated inactivation has been reported to occur in vitro.

The authors now present Fig 3d, as evidence that ScFdc is inactivated by oxygen. It is unclear what this refers to (ie pure protein/in vitro or is it cells containing ScFdc/in vivo)?

I suspect the latter, which could be consequence of a whole range of issues:

- 1) in vivo production of prFMN-OOH which does inhibit Fdc (see ref 34)
- 2) mechanism based inhibition, oxygen somehow inhibiting the ScFdc (mutants or WT) during turnover
- 3) light mediated inhibition (has light been excluded?)

I again must repeat my initial comments that Fdc is easily purified from *E coli* and has not been found to be particularly difficult to study. Therefore, my original request to delve a bit deeper into the actual apparent oxygen-inactivation through in vitro studies of the evolved variants stands.

Response 1-1:

Thank you for your comment.

As you understand, the results regarding oxygen sensitivity of Fdc1 in the previous revision were conducted with the whole cells containing ScFDC F397H:I398Q (the best Fdc1 mutant in this study). All of the experiments with Fdc1 in this manuscript were conducted under light-excluded conditions to prevent the inactivation of Fdc1 by the light.

As you pointed out, we did not show the results regarding oxygen sensitivity of purified Fdc1. We thus conducted an *in vitro* experiment with a purified ScFDC F397H:I398Q, and clarified the direct sensitivity of oxygen for it.

The hexa histidine-tag sequence was fused to ScFDC F397H:I398Q for purification and His-tagged ScFDC F397H:I398Q and UbiX were co-expressed in *E. coli* under oxygen-depleted conditions. The ScFDC F397H:I398Q was purified using Ni-NTA column and the purified ScFDC F397H:I398Q was desalted using PD-10 column. The prepared enzyme assay solutions (1 μ M ScFDC F397H:I398Q, 50 mM potassium phosphate buffer consisting of 50 mM potassium chloride) without *cis,cis*-muconic acid (ccMA) were incubated for 0, 10, 20, 30, 45, 60, 120, 180 min under aerobic or oxygen-depleted conditions. After incubation, the substrate ccMA was added. After packing and incubation for 18 h at 37°C, we analyzed the produced 1,3-butadiene in the gas phase of the vial via HS/GC-MS. The activity of ScFDC F397H:I398Q at 0 min was defined as 100%. As a result, the butadiene production decreased as incubation time increased under aerobic conditions, and the yield decreased to 10.1 \pm 1.7% after 180 min. On the other hand, the activity of ScFDC F397H:I398Q was remained (79.6 \pm 2.9%) after 180 min under oxygen-depleted conditions. This experiment was conducted under light-excluded conditions. These results indicate that the activity of ScFDC F397H:I398Q was decreased under aerobic conditions. To produce butadiene from ccMA, the two decarboxylation reactions by ScFDC F397H:I398Q were required, thus the activity of butadiene production may decrease drastically. As you mentioned, the decrease of the FDC activity under aerobic conditions was not reported, so perhaps the conformation change of ScFDC F397H:I398Q by mutations in this study may affect the O₂ sensitivity for it.

Although the mechanism of its inactivation was unclear, the butadiene production catalyzed by ScFDC F397H:I398Q was decreased under aerobic conditions.

We replaced the result of the whole cell activity with that of purified ScFDC F397H:I398Q and revised the manuscript. We added the explanation that all of experiments with FDC were conducted under light-excluded conditions.

While AroY, which uses the same coenzyme that prFMN uses, requires exposure to oxygen to induce activity, it is known that overexposure to oxygen causes loss of enzyme activity²⁹. Therefore, we also investigated the effects of oxygen on ScFDC F397H:I398Q. After incubation of ScFDC F397H:I398Q under aerobic or oxygen-depleted conditions, the ccMA was added and its enzyme activity was measured. The relationship between incubation time and the enzymatic activity of FDC is shown in Fig. 3d. The enzymatic activity of ScFDC F397H:I398Q at 0 min after incubation started was defined as 100%. The ScFDC F397H:I398Q activity has a half-life of 30 min under aerobic conditions but

decreased to $10.1 \pm 1.7\%$ after 180 min. On the other hands, the $79.6 \pm 2.9\%$ of the activity was remained after 180 min under oxygen-depleted conditions. These results indicated that the activity of FDC was decrease due to continued oxygen exposure.

(Please, see page 10, line 162-page11, line 172)

Fig. 3: Design of ScFDC for 1,3-butadiene production.

d Time course of activity of ScFDC F397H:I398Q under aerobic (blue) or oxygen-depleted (red) conditions. The activity was defined as 100% at 0 min after the incubation started. The data are presented as the means \pm SDs of three independent experiments ($n = 3$). ScFDC, ferulic acid decarboxylase derived from *Saccharomyces cerevisiae*. SD, standard deviation.

The enzymatic activity of AroY requires oxygen to form an active prFMN; however, increased oxygen exposure leads to decrease the activity²⁹. Additionally, it was recently reported that oxidative maturation of prFMN is required for FDC activity.³⁴ The oxygen sensitivity of ScFDC F397H:I398Q was measured, and decrease of enzymatic activity of FDC under aerobic conditions was observed. To produce butadiene from ccMA, the two decarboxylation reactions by ScFDC F397H:I398Q were required, thus the activity of butadiene production may decrease drastically. Perhaps, the conformation change of ScFDC F397H:I398Q by mutations in this study may affect the O₂ sensitivity for it.

(Please, see page 20, line 328-336)

We constructed **pET-T7-ScFDC F397H:I398Q-His_tag** as follows: A fragment of the ScFDC F397H:I398Q was amplified via PCR using pET-T7-ScFDC F397H:I398Q as a template in conjunction with the primer pair ScFDC His-tag_fw and ScFDC His-tag_rv. The fragment was conjugated and the obtained plasmid was named pET-T7-ScFDC F397H:I398Q-His_tag.

(Please, see page 30, line 500-504)

For analyzing the time course of activity of ScFDC F397H:I398Q, hexa histidine-tagged ScFDC F397H:I398Q and UbiX was also coexpressed in *E. coli*. ScFDC F397H:I398Q was purified with a His-tag attached to the C-terminal of ScFDC F397H:I398Q using a Ni-NTA column (His-Trap HP column 5 mL, GE Healthcare Bio-Sciences Uppsala, Sweden) in 50 mM phosphate buffer, 50 mM KCl, pH 7, with wash and elution buffers supplemented with 10 and 250 mM imidazole, respectively. Finally, purified His-tagged ScFDC F397H:I398Q was desalted into 20 mM phosphate buffer containing 50 mM KCl (pH 6.0) on PD 10 Sepharose columns (GE Healthcare Bio-Sciences Uppsala, Sweden).

The time course of activity of ScFDC F397H:I398Q was analyzed. 0.1 vvm (volume of gas per volume of liquid per minute) of nitrogen was bubbled through the 100 mM potassium phosphate buffer consisting of 100 mM KCl (pH 6.0) for 30 min. Then, the enzyme assay solution containing 1 μ M ScFDC F397H:I398Q, 50 mM potassium phosphate buffer consisting of 50 mM KCl (pH 6.0) was prepared. The solutions were incubated 0, 10, 20, 30, 45, 60, 120, 180 min under aerobic or oxygen-depleted conditions. After incubation, 8 mL of the enzyme assay solution was transferred into a 10-mL HS/GC-MS vial, and 8 μ L of 500 mM disodium ccMA stock was added. The final concentration of ccMA was 0.5 mM. After packing and incubation for 18 h at 37°C, we analyzed the produced 1,3-butadiene in the gas phase of the vial via HS/GC-MS. The activity of ScFDC F397H:I398Q was calculated on the basis of the amount of produced butadiene.

(Please, see page 36, line 599-page37, line 616)

All of the experiments with FDC were conducted under light-excluded conditions to prevent the inactivation of FDC by the light.²⁷

(Please, see page 38, line 628-629)

Comment 1-2:

My second main comment is about good reporting of values in the text of the main manuscript, which are frequently (if any!) reported without error, and I suspect almost all with precision that are beyond the accuracy of the measurement. This should be checked throughout and corrected. As an example see: line 186, top of p12: 56.3% ? (this suggest the error is in the range of 0.1%?), top of p13 line 208: 2.17 and 2.52 (this suggest errors of 0.01?) and later 0.134 mol (so +- 0.001 mol?)

Response 1-2:

Thank you for your advice and we sincerely apologize the lack of precise notation about the error. We have checked the all of values in the text of the main manuscript and added the error.

REVIEWERS' COMMENTS

Reviewer #1 (Remarks to the Author):

This manuscript is now much improved, I have just minor cosmetic changes to request:

A) Fig 1, legend, [O] overexposure to oxygen is not clear, as it refers to both the oxidative maturation of prFMN as well as the here reported inactivation by O₂ of the evolved variant.

B) P20 suggested changes to new text inserted

"The enzymatic activity of UbiD species requires oxygen to form the active prFMNiminium species [refer to any recent UbiD review here] however, increased oxygen exposure leads to decrease the activity of some Ibid-Family members such as AroY 29. It is reported that oxidative maturation of prFMN is required for FDC activity 34, although no oxygen sensitivity of the active holo-enzyme has been reported. The oxygen sensitivity of purified ScFDC F397H:I398Q was measured in vitro, and a decrease of enzymatic activity of FDC under aerobic conditions was observed. To produce butadiene from ccMA, the two decarboxylation reactions by ScFDC F397H:I398Q are required, thus butadiene production may decrease drastically under aerobic conditions. Perhaps, the conformational change induced by ScFDC F397H:I398Q mutations in this study may lead to O₂ sensitivity not observed for the WT enzyme."

C) Some recent publications should ideally be included in the text as they are relevant to the topic of hydrocarbon production by UbiD/Fdc:

<https://pubs.acs.org/doi/10.1021/acssynbio.0c00464>

the most recent review on UbiD enzymes:

DOI: 10.1016/bs.enz.2020.05.013

Responses to comments by Reviewer #1:

Thank you for your comments that have helped us to improve our manuscript. We revised our manuscript according your suggestion as follows.

Comment 1-1:

A) Fig 1, legend, [O] overexposure to oxygen is not clear, as it refers to both the oxidative maturation of prFMN as well as the here reported inactivation by O₂ of the evolved variant.

Response 1-1:

We added the explanations “oxidative maturation” and “overexposure” in Fig. 1.

Comment 1-2:

B) P20 suggested changes to new text inserted

"The enzymatic activity of UbiD species requires oxygen to form the active prFMNiminium species [refer to any recent UbiD review here] however, increased oxygen exposure leads to decrease the activity of some Ibid-Family members such as AroY 29. It is reported that oxidative maturation of prFMN is required for FDC activity 34, although no oxygen sensitivity of the active holo-enzyme has been reported. The oxygen sensitivity of purified ScFDC F397H:I398Q was measured in vitro, and a decrease of enzymatic activity of FDC under aerobic conditions was observed. To produce butadiene from ccMA, the two decarboxylation reactions by ScFDC F397H:I398Q are required, thus butadiene production may decrease drastically under aerobic conditions. Perhaps, the conformational change induced by ScFDC F397H:I398Q mutations in this study may lead to O₂ sensitivity not observed for the WT enzyme."

Response 1-2:

As you suggested, we inserted the above text in the manuscript.

(Please, see page 20)

Comment 1-3:

C) Some recent publications should ideally be included in the text as they are relevant to the topic of hydrocarbon production by UbiD/Fdc:

<https://pubs.acs.org/doi/10.1021/acssynbio.0c00464>

the most recent review on UbiD enzymes:

DOI: 10.1016/bs.enz.2020.05.013

Response 1-3:

We cited these articles as reference.

35. Saaret, A., Balaikaite, A. & Leys, D. Biochemistry of prenylated-FMN enzymes. Flavin-Dependent Enzymes. *Enzymes* **47**, 517–549 (2020).
36. Messiha, H. L., Payne, K. A. P., Scrutton, N. S. & Leys, D. A Biological Route to Conjugated Alkenes: Microbial Production of Hepta-1,3,5-triene. *ACS Synthetic Biology* **10**, 228–235 (2021).